# Impaired HA-specific T follicular helper cell and antibody responses to influenza vaccination are linked to inflammation in humans

**Danika L Hill[1,2]\*, Carly E Whyte[2], Silvia Innocentin[2], Jia Le Lee[2], James Dooley[2], Jiong Wang[3], Eddie A James[4], James C Lee[5,6], William W Kwok[7,8], Martin S Zand[3], Adrian Liston[2], Edward J Carr[2,5], Michelle A Linterman[2]\***

[1]Department of Immunology and Pathology, Monash University, Melbourne, Australia; [2]Immunology Program, The Babraham Institute, Babraham Research Campus, Cambridge, United Kingdom; [3]Division of Nephrology, Department of Medicine and Clinical and Translational Science Institute, University of Rochester Medical Center, Rochester, United States; [4]Benaroya Research Institute at Virginia Mason, Translational Research Program and Tetramer Core Laboratory, Seattle, United States; [5]Department of Medicine, Cambridge Biomedical Campus, University of Cambridge, Cambridge, United Kingdom; [6]Cambridge Institute of Therapeutic Immunology & Infectious Disease, Jeffrey Cheah Biomedical Centre, Cambridge Biomedical Campus, University of Cambridge, Cambridge, United Kingdom; [7]Benaroya Research Institute at Virginia Mason, Diabetes Program, Seattle, United States; [8]Department of Medicine, University of Washington, Seattle, United States

**\*For correspondence:**
danika.hill@monash.edu (DLH);
Michelle.Linterman@babraham.ac.uk (MAL)

**Competing interest:** The authors declare that no competing interests exist.

**Abstract** Antibody production following vaccination can provide protective immunity to subsequent infection by pathogens such as influenza viruses. However, circumstances where antibody formation is impaired after vaccination, such as in older people, require us to better understand the cellular and molecular mechanisms that underpin successful vaccination in order to improve vaccine design for at-risk groups. Here, by studying the breadth of anti-haemagglutinin (HA) IgG, serum cytokines, and B and T cell responses by flow cytometry before and after influenza vaccination, we show that formation of circulating T follicular helper (cTfh) cells was associated with high-titre antibody responses. Using Major Histocompatability Complex (MHC) class II tetramers, we demonstrate that HA-specific cTfh cells can derive from pre-existing memory CD4$^+$ T cells and have a diverse T cell receptor (TCR) repertoire. In older people, the differentiation of HA-specific cells into cTfh cells was impaired. This age-dependent defect in cTfh cell formation was not due to a contraction of the TCR repertoire, but rather was linked with an increased inflammatory gene signature in cTfh cells. Together, this suggests that strategies that temporarily dampen inflammation at the time of vaccination may be a viable strategy to boost optimal antibody generation upon immunisation of older people.

## Introduction

Vaccination is an excellent intervention to limit the morbidity and mortality caused by infectious disease. Yet, despite their success, most vaccines are not completely effective, and efficacy varies significantly between different vaccines. The seasonal influenza vaccine needs to be administered

each year in order to provide protection against the most prevalent circulating influenza strains, but its efficacy typically ranges from 40 to 80% even when the vaccine is antigenically matched to circulating viruses. This inefficacy contributes to millions of severe influenza cases and hundreds of thousands of deaths globally (*Iuliano et al., 2018*), which could be potentially prevented by a more effective vaccine.

The reasons that the seasonal influenza vaccine provides protection in some individuals, but not others, have yet to be fully established. Antibodies against the influenza surface glycoprotein haemagglutinin (HA) are capable of limiting infection, and anti-HA antibody titres and inhibitory activity are the most commonly used correlate of protection (*Hobson et al., 1972*). The human antibody response to influenza vaccination is highly variable, but what causes this inter-individual variation is not well understood. Twin studies estimate that genetics can account for less than 20% of the variation in antibody responses to influenza vaccination, implicating non-heritable factors as key contributing influences (*Brodin et al., 2015*). Age, sex, chronic viral infections, and non-communicable diseases have all been reported to influence antibody titre following vaccination (*Frost, 2020*; *Furman et al., 2015*; *Giefing-Kröll et al., 2015*; *Hill et al., 2020*; *Nakaya et al., 2015*; *Stebegg et al., 2020*), but how these various factors impact immune responses to vaccination have yet to be fully unravelled.

The generation of protective humoral immunity is supported by CD4$^+$ helper T cells (*Vinuesa et al., 2016*), which, like neutralising antibodies, are correlates of protection for influenza infection (*Wilkinson et al., 2012*). The majority of work on human T cell responses to influenza vaccination has focussed on T helper type 1 cells, largely because of the relative ease of detecting antigen-specific cytokine-secreting cells upon ex vivo peptide restimulation. However, this approach fails to identify T cell types that do not readily secrete cytokines, such as T follicular helper (Tfh) cells (*Dan et al., 2016*; *Havenar-Daughton et al., 2016*), and therefore our understanding of how the human CD4$^+$ T cell response is linked with high-titre antibody responses upon vaccination is limited. Here, we use Major Histocompatibility Complex (MHC) class II tetramers (*Uchtenhagen et al., 2016*; *Yang et al., 2013*) to directly assess helper T cell responses to the seasonal influenza vaccine. We find that differentiation of HA-specific circulating T follicular helper (cTfh) cells, but not the total number of HA-specific CD4$^+$ T cells, is correlated with high-titre antibody production upon vaccination. HA-specific cTfh cells are clonally expanded from memory cells present pre-vaccination and share a transcriptional profile with human lymph node Tfh cells. Further, we find that in older people there is a specific defect in the formation of cTfh cells upon vaccination. Interestingly, this was not explained by limited T cell receptor (TCR) diversity of the responding T cells, as is commonly proposed as the cause of poor T cell responses in older people (*Looney et al., 2001*; *Yager et al., 2008*). Rather, poor cTfh cell and antibody responses correlated with an enhanced inflammatory gene signature. Together, this implicates cTfh cells as key mediators of antigen-specific immunity and suggests that vaccine strategies that limit, rather than enhance, the inflammation associated with ageing may be more successful in older individuals.

## Results

### HA-specific CD4$^+$ T cells expand and differentiate in response to seasonal influenza vaccination

In order to track influenza HA-specific CD4 T cells directly ex vivo, we recruited two cohorts of healthy UK individuals between 18 and 36 years old across two Northern Hemisphere influenza seasons in 2014/2015 (n = 16, median 30.5 years old) and 2016/2017 (n = 21, median 25 years old) carrying either *HLADR\*0701* or *HLADR\*1101* alleles. Blood samples were collected at baseline (d0), 7 days (d7), and 42 (d42) days after seasonal trivalent influenza vaccination (TIV). A total of 53 distinct variables were measured at two or more time points (n = 147 total measures, *Figure 1A*, *Supplementary file 1*, *Figure 1—figure supplements 1–3*). IgG levels increased against all measured HA proteins from the vaccine influenza strains at d7 and d42 (*Figure 1B*). We analysed post-vaccination time points (d7 and d42) compared to baseline to identify which immunological parameters were altered by vaccination (*Figure 1C*). From our panel of 32 HA proteins, IgG titres to 31 (96%) were altered at d42 relative to baseline (d0), with the greatest fold changes observed for HA strains contained in the TIV. These data indicate that vaccine-induced IgG responses were able to partially cross-react across multiple influenza strains (*Figure 1C*). The increase in anti-HA antibody titre was coupled with an increase

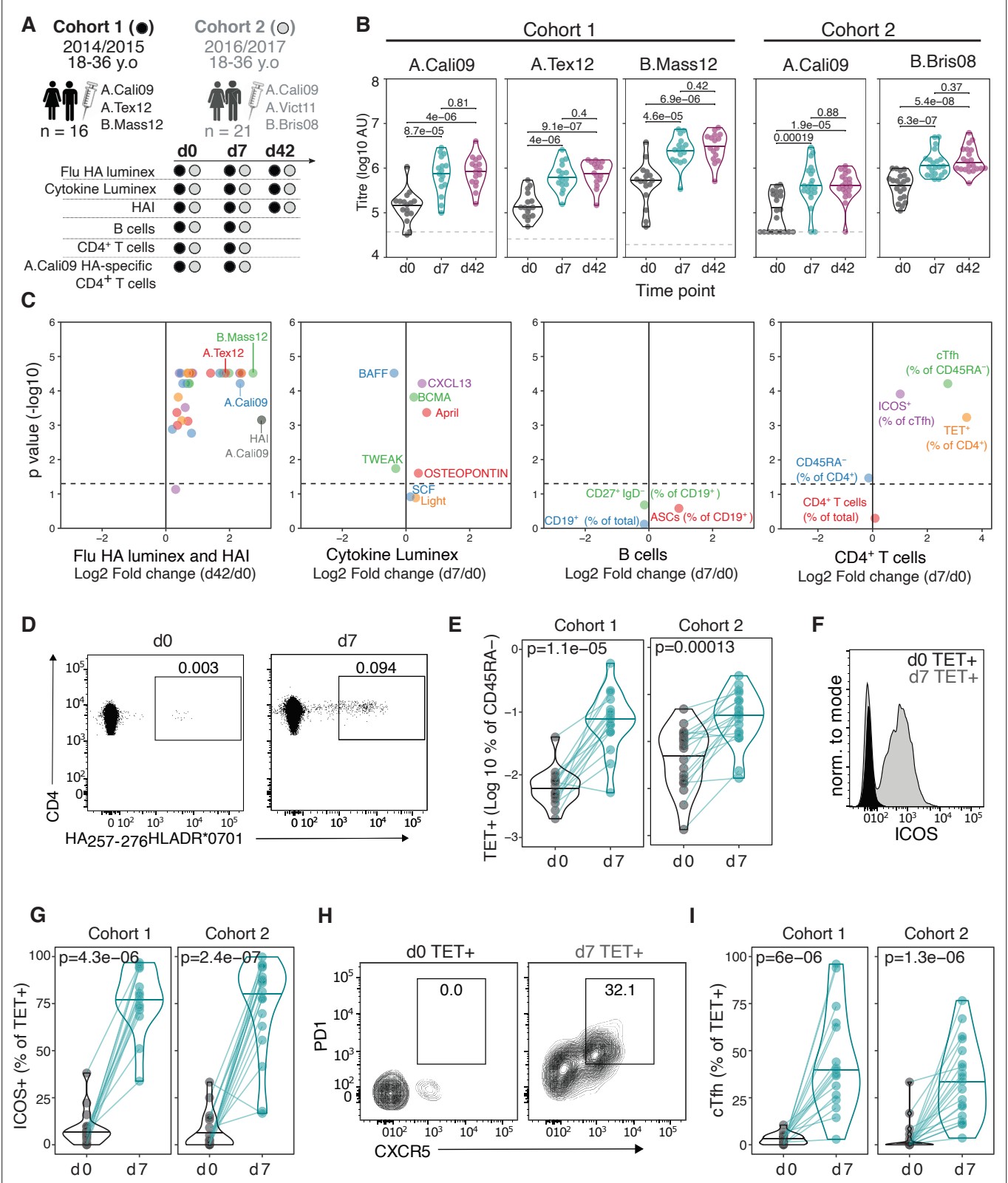

**Figure 1.** Robust haemagglutinin (HA)-specific CD4+ T cells' response to seasonal influenza vaccination. (**A**) Overview of cohort characteristics, vaccination strains, and immune variables measured before (d0), 7 days (d7), and 42 days (d42) after seasonal influenza vaccination. (**B**) IgG responses to HA proteins from vaccine influenza strains measured by Luminex. Dashed line indicates limit of detection in Luminex assay. (**C**) Log2 fold change versus -log10 false discovery rate (FDR)-adjusted p-value of Flu HA Luminex, Cytokine Luminex, B cell and CD4+ T cell immunophenotyping parameters

*Figure 1 continued on next page*

*Figure 1 continued*

before and after vaccination at indicated time points for cohort 1. Dashed line represents the p-value cut-off at FDR-adjusted p=0.05. (**D**) Representative flow cytometry plots of HA$_{257-276}$HLADR*0701 staining on CD4$^+$CD45RA$^-$ cells, gate corresponds to HA-specific TET$^+$ T cells, and (**E**) the frequency of HA-specific TET$^+$ T cells among all CD4$^+$CD45RA$^-$ cells on d0 compared to d7. (**F**) Representative ICOS staining and (**G**) the percentage of ICOS$^+$ cells among HA-specific TET$^+$ T cells on d0 and d7. (**H**) CXCR5 and PD1 staining on HA-specific TET$^+$ T cells on d0 and d7, gate corresponds to 'TET$^+$cTfh' T cells, and (**I**) the percentage of circulating T follicular helper (cTfh) cells among HA-specific TET$^+$ T cells on d0 and d7 for each cohort. In parts (**D–I**), cohort 1 n = 16; cohort 2 n = 19. Paired p-values determined using Wilcoxon signed-rank test.

The online version of this article includes the following figure supplement(s) for figure 1:

**Figure supplement 1.** B cell flow cytometry gating strategy.

**Figure supplement 2.** CD4$^+$ T cell flow cytometry gating strategy.

**Figure supplement 3.** Haemagglutinin (HA)-specific CD4$^+$ T cell sorting strategy.

**Figure supplement 4.** Haemagglutination inhibition (HAI) responses before and after vaccination.

**Figure supplement 5.** Cytokine and CD4$^+$ T cell variables altered after vaccination in 18–36-year-old individuals.

**Figure supplement 6.** Percentage of circulating T follicular helper (cTfh) cells that are Tet$^+$ and CXCR3 and CCR6 expression on haemagglutinin (HA)-specific CD4$^+$ T cells.

in haemagglutination inhibitory antibodies to A.Cali09, the one influenza A strain contained in the TIVs that was shared across the two cohorts and showed a positive correlation with the A.Cali09 IgG titres measured by Luminex assay (*Figure 1C*, *Figure 1—figure supplement 4*). Our analysis of eight cytokines by Luminex identified that four cytokines were upregulated (CXCL13, BCMA, APRIL, and Osteopontin) and two were downregulated (BAFF and TWEAK) at d7 post-vaccination (*Figure 1C*, *Figure 1—figure supplement 4*). We did not detect alterations in the frequency of any B cell subsets in either cohort by flow cytometry; however, the frequency of cTfh cells (CD45RA$^-$CXCR5$^+$PD1$^{++}$) was increased on d7 (*Figure 1C*, *Figure 1—figure supplement 5*), a population that we and others have previously shown share transcriptional and clonal similarity with germinal centre Tfh cells (*Brenna et al., 2020*; *Heit et al., 2017*; *Hill et al., 2019*; *Locci et al., 2013*). Furthermore, the expression of ICOS on cTfh cells was increased on d7, confirming the cTfh population is an activated effector population that forms in response to vaccination (*Figure 1C*, *Figure 1—figure supplement 5*).

To study HA-specific CD4$^+$ T cell responses, we focussed our analysis on the A.Cali09 strain as this was included in both seasons' vaccine formulations, and used MHC class II tetramers of HLADR*0701 and HLADR*1101 loaded with unique A.Cali09 HA peptides to identify antigen-specific T cells (*Yang et al., 2013*). Tetramer-binding antigen-experienced CD4$^+$CD45RA$^-$ (Tet$^+$) T cells showed the largest fold change increase after vaccination of any parameter measured (*Figure 1C*). Tet$^+$ cells were detected in all individuals before vaccination and expanded a median of 5–11-fold between d0 and d7 in both cohorts (*Figure 1D and E*). These antigen-specific T cells had upregulated ICOS after immunisation, indicating that they have been activated by vaccination (*Figure 1F and G*). In addition, a median of one third of HA-specific T cells upregulated the Tfh markers CXCR5 and PD1 on d7 after immunisation (*Figure 1H* and I). The tetramer-binding cells represented between 0.022 and 2.7% of the total CXCR5$^+$PD-1$^+$ bulk population (*Figure 1—figure supplement 6A*, B). The majority of Tet$^+$ cTfh cells expressed CXCR3 (*Figure 1—figure supplement 6C* and D), consistent with the 'Th1' skew in the total CD4$^+$ T cell (*Yang et al., 2013*) and cTfh cell response to influenza vaccination (*Bentebibel et al., 2013*). Therefore, seasonal influenza vaccination increases anti-HA IgG titres, induces a cTfh response, and promotes the expansion and differentiation of HA-specific CD4$^+$ T cells.

## Circulating HA-specific Tfh cells correlate with vaccine IgG response

The majority of successful vaccines provide protection against re-infection through the production of antibodies. The development of antibody secreting plasma cells requires a concerted effort of multiple cell types of the immune system, which have been investigated in detail in mice, but not well in humans. Therefore, we sought to determine which immune parameters were linked with A.Cali09 IgG production 6 weeks post-vaccination (*Figure 2A–D*). Pre-existing antibody titres have been linked with diminished responses to subsequent vaccination (*Tsang et al., 2014*; *Sasaki et al., 2008*; *Bucasas et al., 2011*), but while we observed a slight negative correlation at d0 in support of this, the relationship was not statistically significant for any of the Flu HA Luminex or A.Cali09 HAI titres (*Figure 2B*). In contrast, the IgG responses to a range of HA proteins from different influenza strains correlated

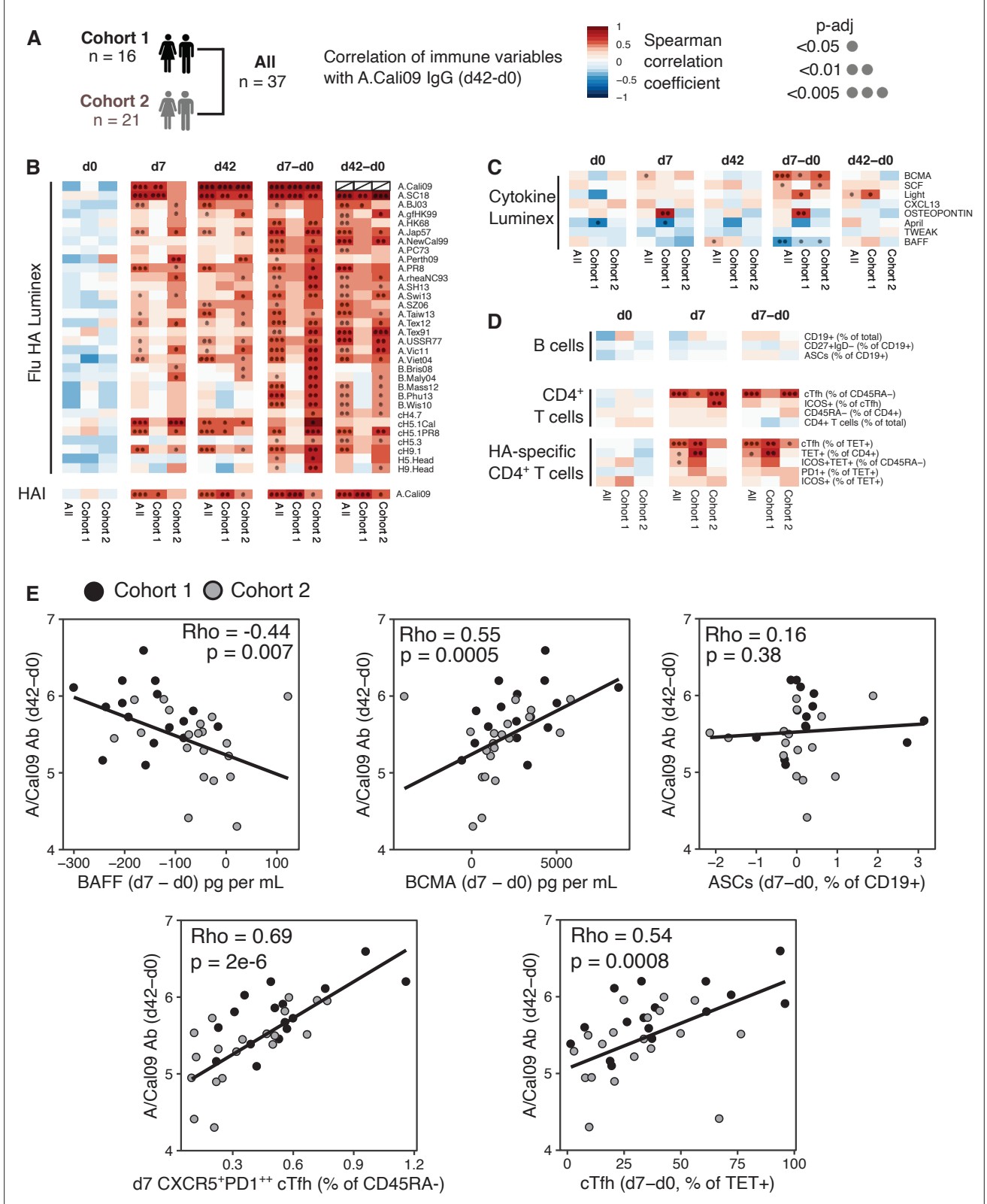

**Figure 2.** Circulating haemagglutinin (HA)-specific T follicular helper (Tfh) cells correlate with vaccine IgG response. (**A**) Overview of correlation analysis between A.Cali09 IgG response at day 42 (minus d0 baseline titre, d42-d0) and immune variables for cohort 1, cohort 2, and both cohorts combined (All). (**B**) Correlations for Flu HA Luminex IgG and haemagglutination inhibition (HAI) titres at d0, d7, and d42, and at d7 and d42 after subtracting each individual's d0 baseline value (d7-d0, d42-d0). (**C**) Correlations for serum cytokines measured by Luminex at d0, d7, and d42, and at d7 and d42 after

*Figure 2 continued on next page*

*Figure 2 continued*

subtracting each individual's d0 baseline value (d7-d0, d42-d0). (**D**) Correlations for B cell, CD4⁺ T cell, and HA-specific CD4⁺ T cell variables at d0, d7, and d7 after subtracting each individual's d0 baseline value (d7-d0). (**E**) Correlation between vaccine-induced A.Cali09 IgG at d42 with selected immune parameters in both cohort 1 and cohort 2 (n = 37). Dot colour corresponds to the cohort (black = cohort 1, grey = cohort 2). Coefficient (Rho) and p-value determined using Spearman's correlation, and line represents linear regression fit.

The online version of this article includes the following figure supplement(s) for figure 2:

**Figure supplement 1.** Correlations between haemagglutination inhibition (HAI) assay titres and selected immune parameters.

strongly with A.Cali09 IgG (d42-d0), indicating that those individuals with large vaccine-induced IgG responses also developed cross-reactive antibodies against multiple strains (*Figure 2B*). Changes between d0 and d7 in serum BCMA were positively correlated, and BAFF negatively correlated with A.Cali09 IgG (d42-d0) responses, in both cohorts (*Figure 2C and E*). The frequency of B cell subsets, including antibody-secreting cells at d7, was not associated with d42 A.Cali09 IgG (d42-d0), whereas total cTfh cell frequency correlated with antibodies in both cohorts (*Figure 2D and E*). A reproducible positive correlation was observed between A.Cali09 IgG (d42-d0) and HA-specific cTfh cells (*Figure 2D and E*), but not consistently for total Tet⁺ CD4⁺ T cells (*Figure 2D*). This indicates that the differentiation of antigen-specific Tfh cells is more relevant for antibody responses than the overall frequency of antigen-specific helper T cells. Similar trends were seen when these immune parameters were correlated to HAI titres against A/Cali09 (*Figure 2—figure supplement 1*).

## HA-specific cTfh response to influenza vaccination includes recalled and public TCR clonotypes

Our results demonstrated that HA-specific cTfh cells are correlated with the antibody response to seasonal influenza vaccination, and so we next investigated how vaccination influences the TCR repertoire and transcriptional signatures of HA-specific cTfh cells and their precursors. We sort-purified and RNA-sequenced tetramer-binding CD4⁺ T cells (Tet⁺ cells) from d0 and Tet⁺ cTfh cells from d7 (*Figure 3A*, *Figure 1—figure supplement 3*, *Figure 3—figure supplement 1*) and retrieved a total of 1405 and 2085 TCRβ clonotypes at d0 and d7, respectively. Expanded clones were observed on d7 (*Figure 3B*), resulting in a decrease in the diversity of the TCR repertoire at d7 relative to d0 (*Figure 3C*) and an increase in the number of co-dominant TCRβ clonotypes within the HA-specific TCR repertoire of each individual (*Figure 3D*). Analysis of paired samples that were sequenced at both d0 and d7 demonstrated that TCRβ clonotypes were recalled from the d0 memory CD4⁺ T cell pool into the cTfh cell response at d7 in 10 out of 15 individuals (*Figure 3E and F*). Recalled TCRβ clones represented a median of 25% of the overall TCR repertoire at d7 (*Figure 3G*) and were more likely to have been more abundant at d0 compared to clones only detected at d0 (*Figure 3H*). Next, we sought to compare pre- and post-vaccination TCR repertoires between individuals. Identical TCRβ sequences between two or more individuals, or 'public clonotypes', were present in 18 out 20 cTfh Tet⁺ samples at d7 and represented a median of 10% of the total response (total of 32 distinct public TCR sequences, *Figure 3I and J*). In contrast, only four public clonotypes were shared across seven individuals at d0. There was a tendency for public clonotypes to preferentially use particular *TCRBV* and *TCRBJ* gene combinations (*Figure 3K*). These results demonstrate that the vaccine-induced cTfh response to A.Cali09 involves reactivation of existing memory CD4⁺ T cells, and that common responses between individuals are a prevalent part of the cTfh cell response.

## Vaccination-induced cTfh cells share a common transcriptional signature with lymph node Tfh cells

We sought to determine the transcriptional profile of HA-specific cTfh (Tet⁺) cells and to what extent these cTfh Tet⁺ cells acquired the transcriptional signature of bona fide human lymph node germinal centre Tfh cells. HA-specific cTfh cells that form 7 days after vaccination clustered distinctly from their d0 Tet⁺ precursors cells by principal component analysis (PCA) for both cohorts (*Figure 4A*). 684 genes were differentially expressed (DE) in Tet⁺ cTfh cells compared to d0 Tet⁺ cells across both cohorts (DESeq2 log₂FC > 0.5, FDR p-value<0.1, *Source data 1*). This gene signature was compared to a 1179 gene signature generated by comparing germinal centre Tfh cells (CXCR5⁺PD1⁺⁺⁺) to non-Tfh antigen-experienced CD4⁺ T cells from human inguinal lymph nodes (*Hill et al., 2019*; *Source*

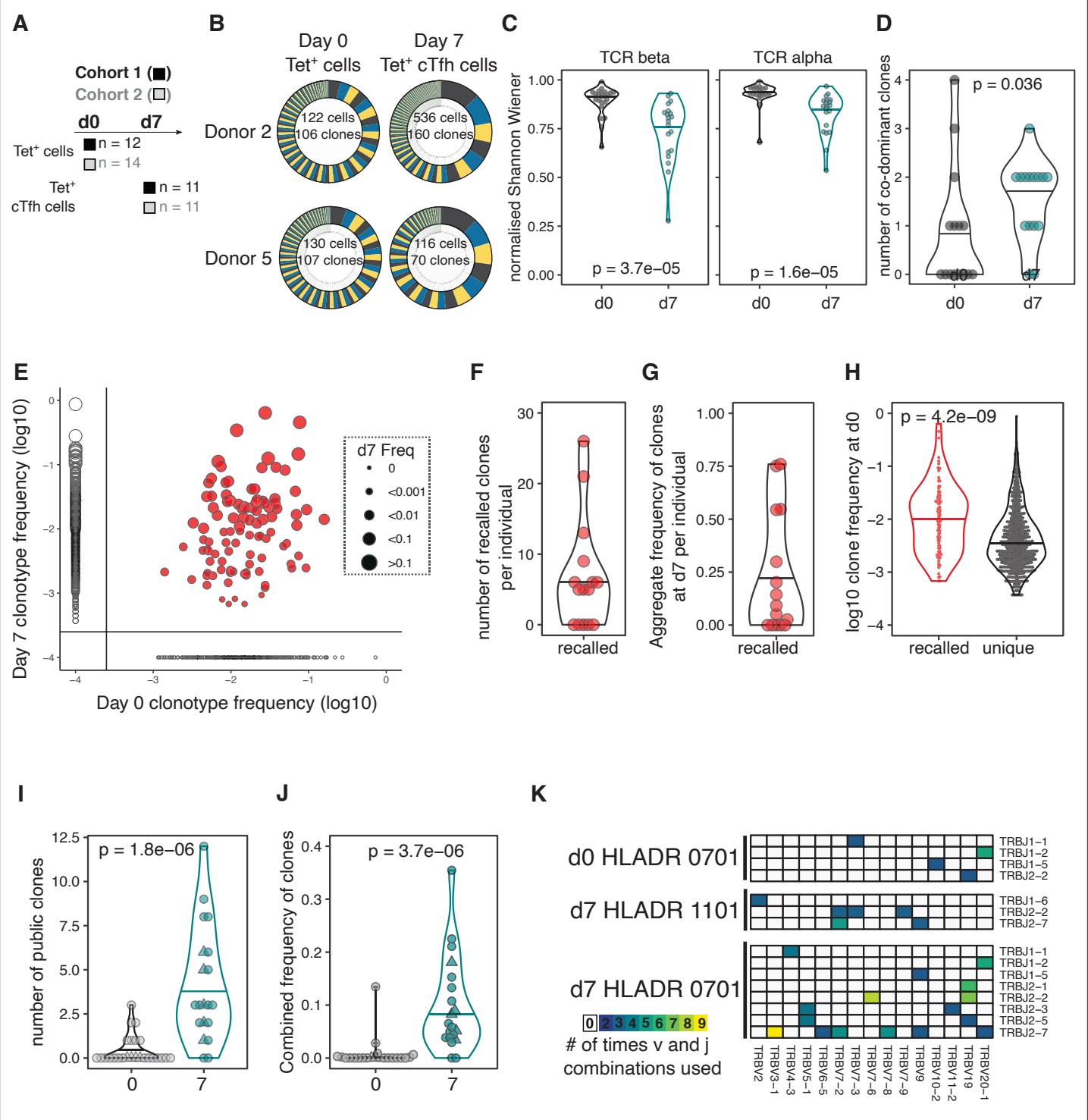

**Figure 3.** Haemagglutinin (HA)-specific circulating T follicular helper (cTfh) response to flu vaccination includes recalled and public T cell receptor (TCR) clonotypes. (**A**) Overview of cell types and sample sizes that were sequenced d0 and d7 at each cohort. d0 median = 45 cells (6–258); d7 median = 88 cells (5–1000). (**B**) Representative pie charts of the proportions of unique TCR $\beta$-chain clonotypes for participants 408S and 425L at d0 and d7. Inset numbers indicate the number of cells sequenced and number of unique TCR $\beta$ clonal sequences retrieved. (**C**) Normalised Shannon–Wiener index of TCR $\beta$ and TCRα repertoire diversity at d0 TET⁺ T cells and d7 TET⁺ cTfh cells for both cohorts combined. (**D**) The number of dominant TCR $\beta$ clones (frequency of >10%) for each individual in d0 and d7 sequenced cells for both cohorts combined. (**E**) TCR $\beta$ clonotype frequencies at d0 and d7 in combined dataset of 15 individuals with paired d0 and d7 samples (1772 clones in total). Each dot represents a clonotype, size corresponds to d7 frequency. Red colour indicates 'recalled' clones that were measured at both time points, and solid lines indicate where clones are only detected at a

*Figure 3 continued on next page*

Figure 3 continued

single time point. (**F**) The number of 'recalled' TCR $\beta$ clones per individual (present at both d0 and d7) and (**G**) the aggregate frequency of d7 TET$^+$ cTfh TCR $\beta$ clones that were recalled for each individual (n = 15). (**H**) The log10 d0 frequency of each 'recalled' or 'unique' (present at d0 only) clone (n = 102 recalled, n = 663 unique clones). (**I**) The number and (**J**) combined frequency of public clonotypes per individual at each time point, cohorts 1 and 2 combined (d0 n = 25; d7 n = 20). Public clonotypes were defined as TCR $\beta$ sequences with identical v, d, j genes and CDR3 amino acid sequence shared between two or more samples from the same time point. (**K**) TRBJ and TRBV gene usage among the 26 HA-specific public clonotypes, separated by HLADR genotype and time point. All p-values were calculated using a Mann–Whitney *U* test.

The online version of this article includes the following figure supplement(s) for figure 3:

**Figure supplement 1.** Cell number and clonotype number from T cell sequencing.

*data 2*). 147 of the DE genes in Tet$^+$ cTfh cells shared the same expression pattern with genes up- or downregulated in germinal centre Tfh signature, termed 'Tfh genes'. These included upregulation of key Tfh cell transcription factors *MAF* and *TOX2*, and downregulation of several cytokine and chemokine receptors such as *IL7R, IL2RA,* and *CCR6* (*Figure 4B and C*, *Source data 3*). The remaining 537 genes, termed 'vaccination genes', were modulated after influenza vaccination but not DE in our human lymph node Tfh cell samples. This list included the downregulation of *CXCR4,* a chemokine receptor that is important for lymph node homing consistent with the blood localisation of the cTfh cells (*Okada et al., 2002*), and upregulation of the interferon-gamma-inducible gene *GBP2*. This gene-level analysis indicated that Tet$^+$ cTfh cells share features of a transcriptional profile with lymph node Tfh cells, while also expressing genes specific either to the Th1-skew in the Tfh response to influenza vaccination or to circulation in the blood.

To gain a clearer understanding of what molecular pathways were modulated during the differentiation into Tfh cells after vaccination at a transcriptional level, we performed gene set enrichment analysis using the Hallmark gene sets (*Figure 4D*; *Liberzon et al., 2015*). Ten gene sets were differentially modulated in d7 Tet$^+$ Tfh cells compared to d0 in both cohorts, eight (80%) of which were also enriched in lymph node germinal centre Tfh cells. These 'Tfh gene sets' included the downregulation of several cytokine pathways including IL-2 signalling, a cytokine known to inhibit Tfh differentiation (*Ballesteros-Tato et al., 2012*; *DiToro et al., 2018*), and the upregulation of oxidative phosphorylation, a metabolic process known to be elevated in Tfh cells (*Figure 4E*; *Dong et al., 2019*; *Ray et al., 2015*). Enrichment for the interferon alpha response and reactive oxygen species pathways was not shared with lymph node Tfh cells and may reflect the acute response to inactivated virus in the vaccine to which the lymph node Tfh cells were not exposed. Overall, our analysis demonstrates that seasonal influenza vaccination recalls antigen-specific memory CD4$^+$ T cells and promotes their differentiation into Tfh cells with similarity to lymph node GC Tfh cells.

## Impaired anti-HA IgG responses in older individuals correlate with failure to fully acquire the immune trajectory of younger individuals

We defined the correlates of influenza vaccine antibody responses in 18–36 year olds to be serum concentrations of BAFF and BCMA, and the frequency of both polyclonal and HA-specific cTfh cells. Next we wanted to investigate how the immune response to seasonal influenza vaccination was impacted in older individuals, a group where the influenza vaccine is less efficacious (*Govaert et al., 1994*; *Jefferson et al., 2007*). We compared our cohorts of 18–36 year olds with that of individuals over 65 years old in the same vaccination years (*Figure 5A*, cohort 1 median 69 years old, cohort 2 median 73.43 years old) and observed that older individuals showed lower HA-specific IgG responses compared to 18–36 year olds for all the vaccine strains measured (*Figure 5B*). To investigate whether ageing broadly influenced immune status pre- and post-vaccination, we applied a diffusion pseudotime algorithm to 23 antibody, cytokine or immunophenotyping variables measured at d0 and d7 for both age groups and cohorts (*Supplementary file 1*). The immune profiles formed a single continuous spectrum, with the pseudotime 'vaccination trajectory' beginning with d0 samples and ending with d7 samples from 18- to 36-year-old samples (*Figure 5C*). There was no pre-vaccination difference in trajectory values between the age groups, indicating that ageing did not impact the baseline status for the immune parameters involved in the vaccination response (*Figure 5D*). In contrast, over 65 year olds failed to progress along the vaccination trajectory to the same extent as their 18–36-year-old counterparts (*Figure 5D*), indicating a failure to appropriately respond to vaccination. 11 immune parameters correlated with the vaccination trajectory values, including antibody responses

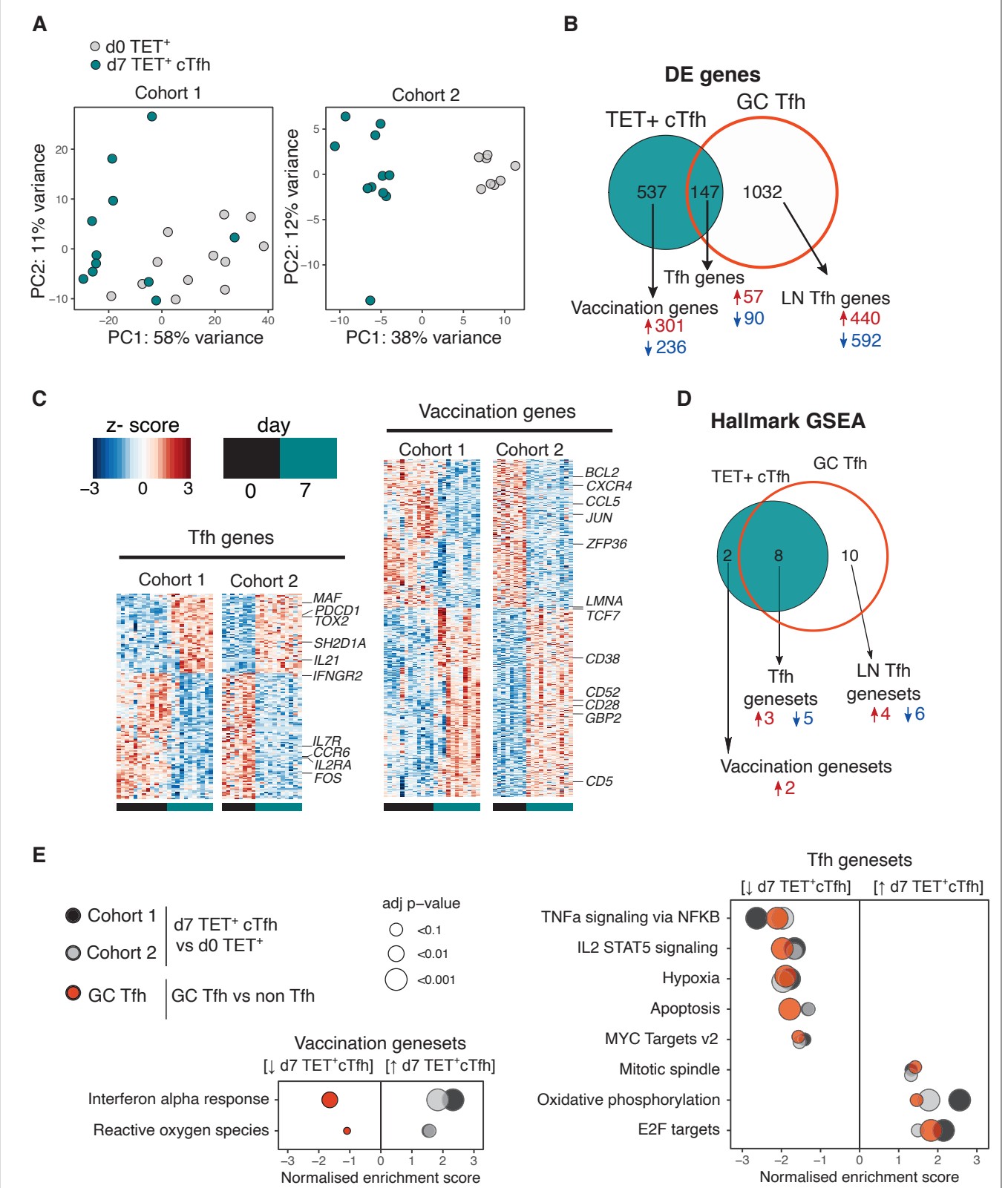

**Figure 4.** Vaccination-induced circulating T follicular helper (Tfh) cells share a common transcriptional signature with lymph node Tfh cells. (**A**) Principal component analysis of 1000 genes with the largest variance in sequenced cells from d0 TET[+] cells (shown in grey) and d7 TET[+] circulating T follicular helper (cTfh) cells (shown in green) for each cohort separately. (**B**) Venn diagram of the 684 significantly differentially expressed (DE) genes in d7 TET[+]cTfh cells (relative to d0) in both cohorts, and the overlap with a signature of human lymph node germinal centre (GC) Tfh cells where

*Figure 4 continued on next page*

*Figure 4 continued*

genes had the same direction of expression. DE genes were determined using DESeq2 and had adjusted p-value<0.1 and fold change of 2. Up- and downregulated genes represented by red and blue colours, respectively. (**C**) Heatmaps of gene signatures determined in (**B**) for each cohort, with representative genes labelled. (**D**) Venn diagram of the 10 consistently enriched Hallmark pathways in d7 TET$^+$cTfh cells relative to d0 TET$^+$ cells in both cohorts, and the overlap with positively or negatively enriched gene sets in LN GC Tfh cells compared to non-Tfh CD4$^+$ cells. Significant pathways were determined using gene set enrichment analysis (GSEA) and had adjusted p-value<0.1 and normalised enrichment score of >1 or <–1. Up- and downregulated pathways are represented by arrows and red and blue colours, respectively. (**E**) Bubble plots showing the normalised enrichment scores for significantly enriched pathways for d7 TET$^+$ cTfh cells versus d0 TET$^+$ cells in each cohort, and LN GC Tfh cells versus non-Tfh cells, with positive and negative scores indicating positive and negative enrichment, respectively, in TET$^+$ Tfh and/or GC Tfh compared to their non-Tfh comparator. Circle colour represents the type of comparison, and size represents the adjusted p-value.

to the vaccination strains, cTfh cells, HA-specific T cells, and diminished serum BAFF (*Figure 5E*). These results indicate that in over 65 year olds there is a failure to fully engage the adaptive immune system in response to vaccination compared to younger individuals.

In order to identify which immune parameters may explain the age-related difference in vaccination trajectory, we compared the 53 immune parameters between age groups before and after vaccination. Older individuals had higher titres of IgG to several HA proteins from different influenza A strains at d0; however, no difference was seen in IgG or HAI titre levels for the influenza strains in the vaccines administered to our cohorts (*Figure 5F*), suggesting that age-dependent differences in pre-existing antibodies to the vaccine strain do not account for the diminished response. After immunisation, over 65 year olds showed lower vaccine-induced IgG responses for numerous HA strains, consistent with the lower IgG responses to the vaccination strains and diminished generation of cross-reactive IgG to other influenza strains (*Figure 5F*). For serum cytokines, we observed consistently lower baseline April and TWEAK concentrations, and higher d42 SCF levels in over 65 year olds across both cohorts (*Figure 5G*, *Figure 5—figure supplement 1A*), but there was no age-dependent difference in BAFF or BCMA. No age-related differences in circulating B cell populations were observed pre- or post-vaccination, and no consistent differences were seen for CD4$^+$ T cell subsets pre-vaccination, including for the frequency of d0 HA-specific CD4$^+$ cells (*Figure 5H*, *Figure 5—figure supplement 1B*). Across both cohorts, the only CD4$^+$ T cell parameters consistently reduced in older individuals at d7 were the frequency of polyclonal cTfh cells and HA-specific Tet$^+$ cTfh cells, with the strongest effect within the antigen-specific cTfh cell compartment (*Figure 5H–J*, *Figure 5—figure supplement 1C*). There was no consistent difference in the total d7 Tet$^+$ HA-specific T cell population with age for both cohorts (*Figure 5H*), and we observed no age-related correlation between the ability of an individual to differentiate Tet$^+$ cells into a cTfh cell and the overall expansion of Tet$^+$ HA-specific T cell population (*Figure 5—figure supplement 1D*). Thus, our data suggest that the poor vaccine antibody responses in older individuals are impacted by impaired cTfh cell differentiation (*Figure 5J*) rather than size of the vaccine-specific CD4$^+$ T cell pool.

## TCR repertoire diversity in HA-specific cTfh cells is comparable in younger and older individuals

Poor T cell responses to vaccination have been previously attributed to a contraction of the naïve CD4$^+$ T cell population and restriction of the TCR repertoire with age (*Looney et al., 2001*; *Yager et al., 2008*), although most human studies have examined the whole repertoire, rather than focusing on vaccine-specific T cells. Our ability to track vaccine-specific CD4$^+$ T cells enables direct assessment of the repertoire of responding cells, and therefore we sought to determine whether there were ageing-related differences in the clonal diversity of pre- or post-vaccination HA-specific CD4$^+$ T cells (*Figure 6A*). The pre-vaccination diversity is relevant as our data from young donors demonstrates that a large contribution to the d7 response originates from memory CD4$^+$ T cells present prior to vaccination. In pre-vaccination HA-specific Tet$^+$ cells, there was a slight reduction in TCR diversity in over 65 year olds compared to younger people, although this was only statistically significant in one cohort (*Figure 6B*). However, after vaccination no difference was seen in the TCR diversity of the resulting d7 Tet$^+$cTfh population between age groups (*Figure 6B*), despite the diminished frequency of this population in older individuals. Likewise, no age-related difference in TCR diversity was observed for the total pool of vaccine-specific Tet$^+$ population at d7 (*Figure 6C*). As we had observed in younger participants, TCRβ clones were present in both the d0 Tet$^+$ and d7 Tet$^+$ cTfh

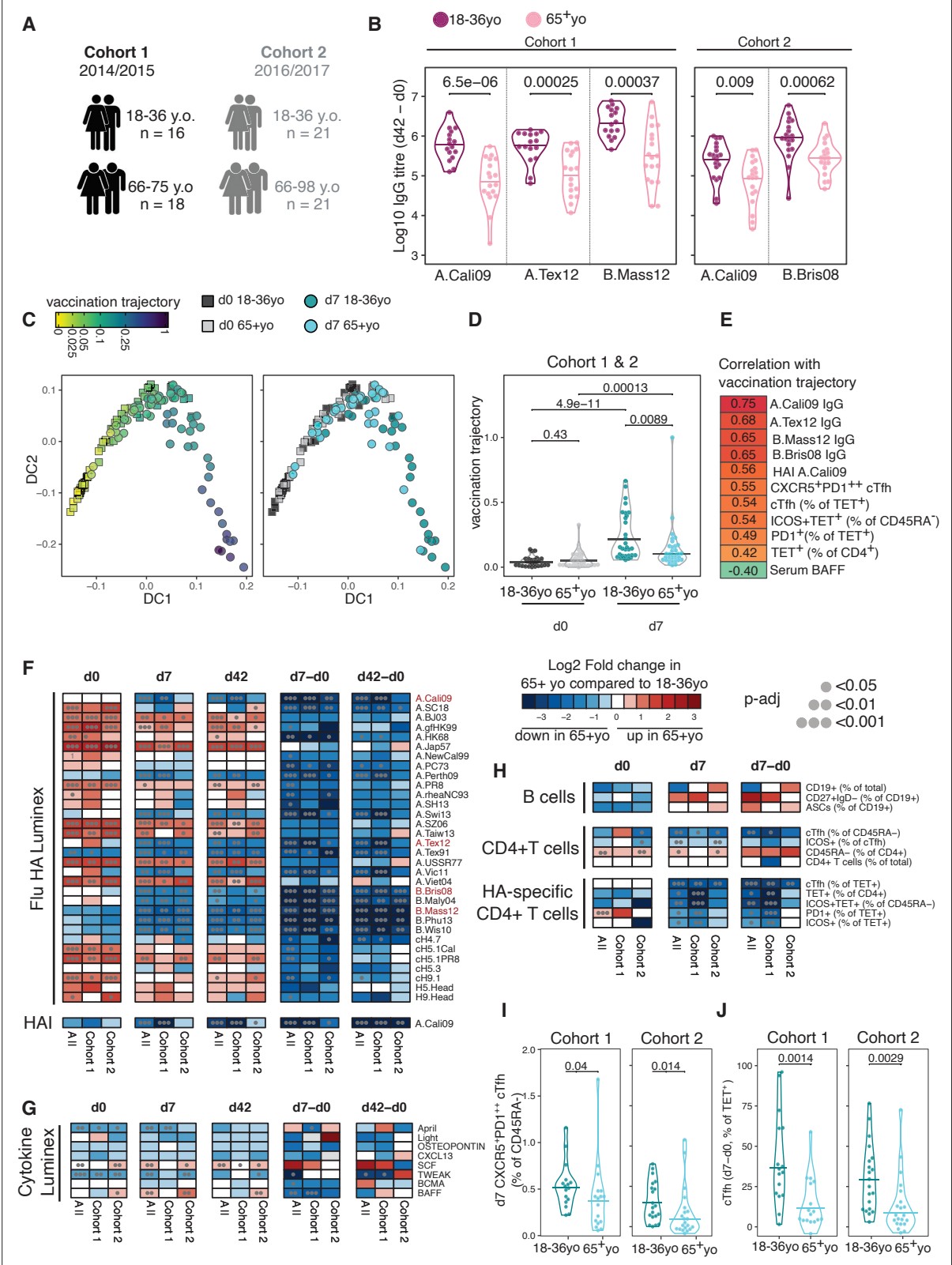

**Figure 5.** Impaired vaccination immune trajectory in older individuals. (**A**) Overview of age groups and sample sizes for each cohort. (**B**) IgG responses to haemagglutinin (HA) proteins from vaccine influenza strains measured by Luminex for each age group. (**C**) Diffusion map dimensionality reduction of 122 samples from both cohorts combined using scaled values for 23 immune parameters and the diffusion-pseudotime algorithm (d0 18–36 year olds n = 27; d0 65+ year olds n = 32; d7 18–36 year olds n = 30; d7 65+ year olds n = 33). Each dot represents a sample, shape represents time point (d0 =

*Figure 5 continued on next page*

*Figure 5 continued*

squares, d7 = circles), and colour either the pseudotime 'vaccination trajectory' output value or age group category. Diffusion components (DCs) 1 and 2 shown. (**D**) Vaccination trajectory values for sample in each age category from both cohorts combined, with p-values calculated using Dunn's post hoc test. (**E**) Spearman's correlation coefficients for the 11 parameters that significantly correlated with the vaccination trajectory variable (padj <0.05). (**F–H**) Heatmap of FDR-adjusted p-values from Mann–Whitney *U* test comparing immune parameters between age groups for cohort 1, cohort 2, and both cohorts combined (All), at d0, d7, and d42, and at d7 and d42 after subtracting each individual's d0 baseline value (d7-d0, d42-d0) for (**F**) Flu HA Luminex and haemagglutination inhibition (HAI), (**G**) Cytokine Luminex, (**H**) B cells, CD4$^+$ T cells, and HA-specific CD4$^+$ T cells. Colour corresponds to p-value and the direction of change. (**I**) The percentage of CXCR5$^+$PD1$^{++}$ circulating T follicular helper (cTfh) cells and (**J**) TET$^+$ cTfh cells for each age group and each cohort, with p-values calculated by Mann–Whitney *U* test (cohort 1 18–36 years old n = 17, 65+ years old n = 17; cohort 2 18–36 years old n = 20, 65+ years o n = 21).

The online version of this article includes the following figure supplement(s) for figure 5:

**Figure supplement 1.** Age-related differences in cytokines and haemagglutinin (HA)-specific CD4$^+$ T cell parameters.

populations from over 65 year olds. There was a trend towards a lower absolute number of recalled clones in older individuals (*Figure 6D*), in line with the lower numbers of d7 sequenced cells due to the reduced cTfh population frequency (*Figure 3—figure supplement 1*). However, the proportion of the d7 population that consisted of recalled clones in each individual was not impacted by age (*Figure 6E*), indicative of equivalent contribution from pre-existing memory cells into the resulting cTfh population between younger and older donors. The affinity of TCR for antigen shapes Tfh differentiation and Tfh versus Th1 cell fate after influenza vaccination (*Tubo and Jenkins, 2014*; *Knowlden and Sant, 2016*; *Keck et al., 2014*), and therefore we sought to determine if the age-related decline in Tet$^+$ cTfh differentiation could be explained by an age-dependent skew in the TCR repertoire away from Tfh differentiation. We examined the TCRβ clones that were present in both the d7 Tet$^+$ and d7 Tet$^+$ cTfh populations and observed strong correlations between clone frequency between the two populations irrespective of age groups, suggesting that there was no restriction in the TCR repertoire able to give rise to Tfh cells to the HA peptides studied here (*Figure 6F*). Together, our data indicate that TCR diversity is not a limiting factor for the Tfh cell response to seasonal influenza vaccination in older individuals and suggest that the age-associated reduction in cTfh cell frequency is due to other cell-intrinsic or environmental effects on Tfh cell differentiation or survival.

## HA-specific cTfh cells from older individuals fail to induce Tfh transcriptional signatures and display aberrant inflammatory signatures

We investigated the transcriptome of HA-specific cTfh cells from older individuals to determine if we could identify pathways that could explain the poor Tfh cell differentiation in ageing. With supervised PC analysis using the 684 genes that are DE in 18–36 year olds' d7 Tet$^+$ cTfh cells compared to d0, we observed that d7 Tet$^+$cTfh cells from over 65 year olds were clustered between d0 and d7 samples from younger donors for PC1 in both cohorts (*Figure 6G*). Of the 425 genes that were consistently DE between HA-specific cells from d0 and d7 in older donors from both cohorts, only 170 (40%) were part of the younger d7 signature (*Source data 5*). Importantly, we did not observe any consistent age-related gene expression differences in pre-vaccination Tet$^+$ d0 cells, indicating that ageing is not associated with transcriptional changes in resting HA-specific memory cells. These data indicate that cTfh cells from older individuals failed to acquire the full gene signature seen in Tfh cells from younger people.

To further resolve these age-related transcriptional differences, we performed enrichment analysis with the Hallmark gene sets and observed seven gene sets that were consistently positively enriched in d7 HA-specific Tet$^+$ cTfh cells from over 65-year-old compared to 18–36-year-old individuals (*Figure 6H*). Four of these elevated gene sets (TNFA signalling via NFKB; IL2 STAT5 signalling; Apoptosis; Hypoxia) we previously identified as negatively enriched in Tet$^+$cTfh from 18- to 36-year-old individuals compared to d0 Tet$^+$ cells. Of these enriched gene sets, all except TNFA signalling via NFKB were also enriched in all Tet$^+$ cells in older persons in addition to cTfh cells, suggesting that there are both common and unique pathways for how vaccine-reactive CD4$^+$ cells and cTfh cells sense their environment in ageing. This indicates that these genes are normally downregulated as HA-specific CD4$^+$ T cells differentiate into Tfh cells in response to vaccination, and that this process appears dysregulated in older individuals. The remaining three enriched gene sets (Interferon gamma response; Inflammatory response; Complement) are suggestive of heightened pro-inflammatory responses in cTfh cells,

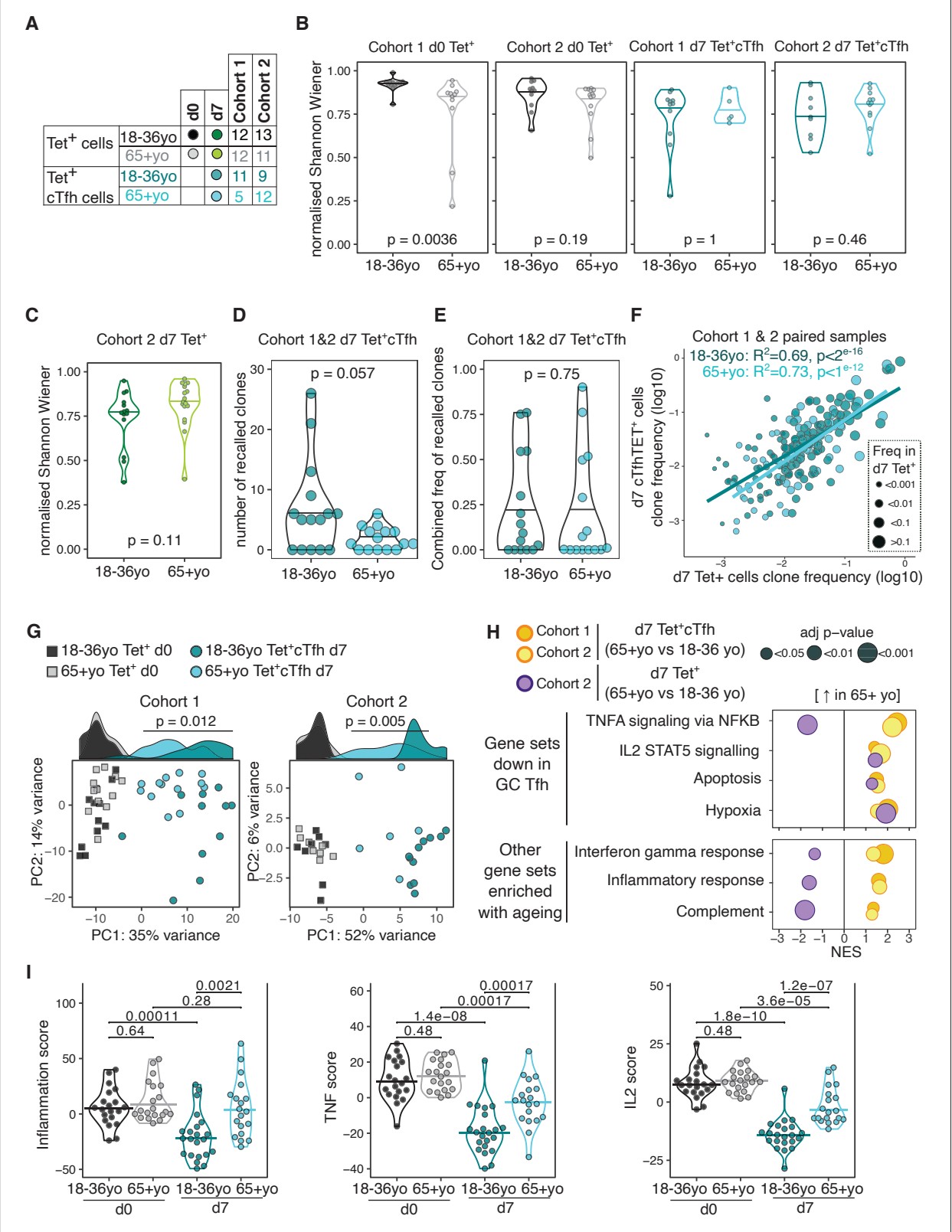

**Figure 6.** Impaired induction of T follicular helper (Tfh) transcriptional programs and aberrant inflammatory signatures in haemagglutinin (HA)-specific circulating T follicular helper (cTfh) cells from older individuals. (**A**) Overview of sample numbers and cell types sequenced at d0 and d7 for each cohort and each age group. Dark and light green dots indicate d7 Tet+ cells sequenced from cohort 2 only. (**B**) Normalised Shannon–Wiener diversity index of T cell receptor $\beta$ (TCR$\beta$) clonotypes for each cohort, time point, and cell type. Each dot represents a sequenced sample. (**C**) Normalised

*Figure 6 continued on next page*

*Figure 6 continued*

Shannon–Wiener diversity index of TCR $\beta$ clonotypes in d7 Tet+ cells from cohort 2. (**D**) The number of TCR $\beta$ clones per individual in d7 Tet+ cTfh cells recalled from d0 Tet+ cells, and (**E**) combined frequency of the recalled clones among d7 Tet+cTfh cells for each age group (n = 15 in each age group, both cohorts combined). p-Values calculated using Mann–Whitney U test. (**F**) Frequency of TCR $\beta$ clones present in paired samples of d7 Tet+ and d7 Tet+cTfh cells. Each dot represents a clonotype, dot colour indicates age group, and dot size corresponds to the frequency in d7 Tet+ cells. Lines represent linear regression, and statistics determined using Pearson's correlation. (**G**) Principal component (PC) analysis of the 684 genes differentially expressed (DE) between d0 and d7 in samples from 18 to 36 year olds applied to d0 and d7 samples from both age groups and cohorts. Histograms show the distribution of PC1 values for each group, with p-value calculated between age groups for d7 samples by Mann–Whitney U test. (**H**) Bubble plots of seven Hallmark pathways that are upregulated in d7 Tet+cTfh cells or d7 Tet + cells in samples from 65+-year-old compared to 18–36-year-old individuals, subdivided by which pathways were previously identified as negatively enriched in lymph node germinal centre Tfh cells. Positive scores indicate positive enrichment in Tet+cTfh cells from older donors. Circle colour represents the cohort, size represents the adjusted p-value. (**I**) Inflammation, TNF, and IL2 gene signature scores in HA-specific T cells at different time points and age groups, cohorts 1 and 2 combined (d0 18–36 year olds n = 20, d0 65+ year olds n = 20; d7 18–36 year olds n = 22; 65+ year olds n = 19). p-Value calculated using Dunn's post hoc test.

but not in all Tet+ CD4+ T cells, from older individuals, and included enrichment for *IL6*, *CCL2*, and *CCL5* (*Figure 6H*). No pathways were altered by ageing in pre-vaccination d0 Tet+ cells, indicating that the enrichment for numerous inflammatory pathways in Tfh cells from older people occurred as part of the response to vaccination, rather than being a generalisable feature of CD4+ biology in ageing. To further investigate the upregulation of inflammation and cytokine signalling in cTfh cells from older donors, we used the leading-edge output from the gene set enrichment results to curate three non-overlapping gene lists of 'TNF', 'IL2', and 'inflammation' (*Supplementary file 2*). We then calculated three gene list scores by generating gene z-scores for each cohort, and then summing the z-scores for the genes in each of the gene lists for each sample, before and after vaccination. For all three gene signatures, while there was no pre-vaccination difference between the age groups, d7 cTfh cells from older donors failed to downregulate the expression of these genes to the same extent as cTfh cells from 18 to 36 year olds (*Figure 6I*). In summary, our transcriptional analysis indicated that in older individuals cTfh cells display evidence of inflammatory cytokine signalling that is not typical of Tfh cells in younger people.

## Vaccine-induced transcriptional signatures of inflammation, TNF, and IL-2 signalling negatively correlate with Tfh differentiation and antibody responses

We sought to determine whether the downregulation of the inflammatory, IL-2, and TNF pathways in Tet+ cTfh cells was linked to Tfh cell differentiation and antibody responses after vaccination. Irrespective of age, the three gene scores were negatively correlated with the Tet+ cTfh cell frequency (*Figure 7A*) and the vaccine-induced A.Cali09 IgG response (*Figure 7B*). Our findings suggest that the presence of pro-inflammatory cytokine signalling in cTfh cells induced by vaccination negatively impacts optimal vaccine humoral responses. In order to determine whether the observed negative correlation between the three inflammatory gene signatures and the IgG response to seasonal influenza vaccination could be replicated in other cohorts, we analysed peripheral blood mononuclear cell (PBMC) microarray data and A.Cali09 anti-HA antibodies from an independent cohort of 50 adults aged 18–86 year olds from the United States (*Nakaya et al., 2015*). Consistent with our cohorts, we did not detect any age-related difference in gene signatures present prior to vaccination, but the gene signatures induced by vaccination d7 (relative to d0) in PBMCs were negatively correlated with IgG production at d28 (*Figure 7C*). This demonstrated that expression of genes associated with inflammation, TNF, and IL-2 signalling in blood is negatively associated with IgG responses to seasonal influenza vaccination, irrespective of age. These data suggest that generation of a robust response to vaccination requires the cytokine milieu in secondary lymphoid tissues to be carefully controlled to limit persistent pro-inflammatory cytokine signalling during Tfh differentiation.

## Sustained IL-2 production inhibits Tfh cell frequency and the germinal centre response

To test the hypothesis that cytokine signalling needs to be curtailed to facilitate Tfh cell differentiation, we turned to a genetically modified mouse model in which cells that have initiated IL-2 production cannot switch it off, *Il2*cre/+; *Rosa26*stop-flox-Il2/+ mice (*Whyte, 2020*). Twelve days after influenza

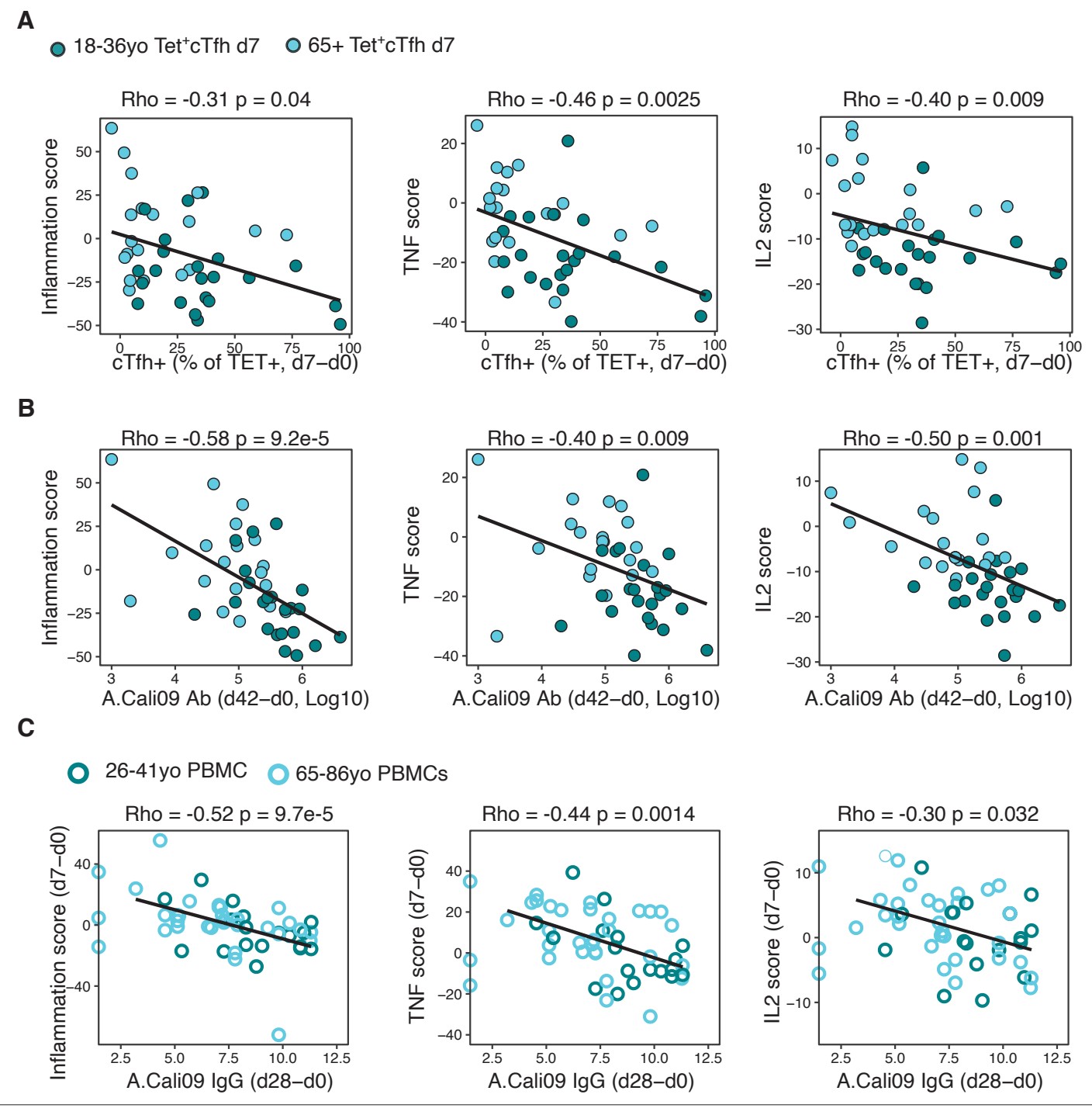

**Figure 7.** Gene signatures of TNF, IL-2, and inflammation associate with impaired antibody and T follicular helper (Tfh) cell responses to influenza vaccination. (**A**) Correlation between circulating T follicular helper (cTfh) Tet+ cells and inflammation, TNF or IL2 gene signatures scores in d7 Tet+cTfh cells. (**B**) Correlation between A.Cali09 IgG titre (d42-d0) and inflammation, TNF or IL2 gene signatures scores in d7 Tet+cTfh cells. (**C**) Correlation between A.Cali09 IgG titre (d28-d0) and Inflammation, TNF or IL2 gene signatures scores determined from microarray data of peripheral blood mononuclear cells on d0 or d7 after seasonal influenza vaccination from publicly available datasets (n = 50 total; 26–41 years old n = 18, 66–86 years old = 32) (**Nakaya et al., 2015**). Correlation coefficients and p-values calculated using Spearman's correlation coefficient. Solid line represents linear regression fit. Colour corresponds to age group (green = younger people; aqua = older people).

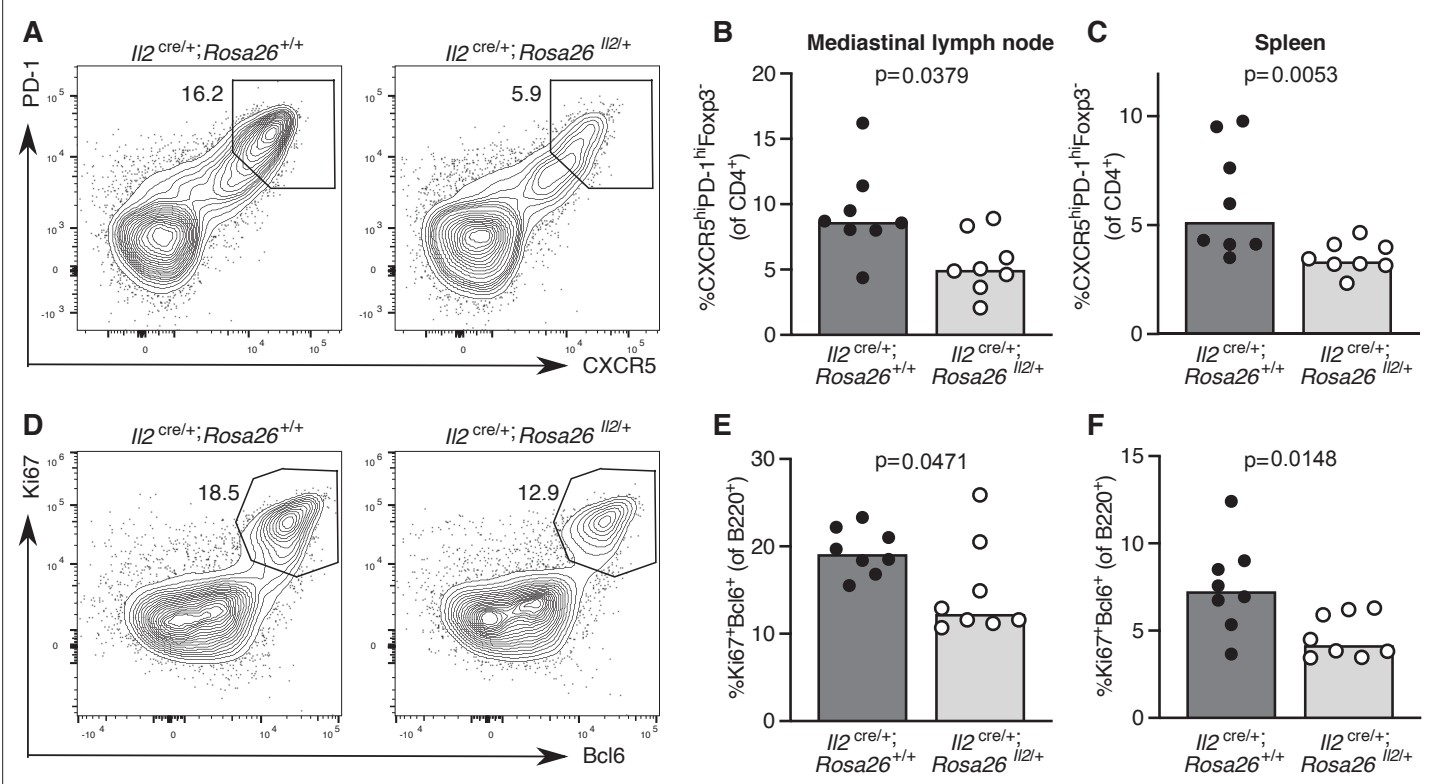

**Figure 8.** Increased IL-2 production impairs T follicular helper (Tfh) cell formation and the germinal centre response. Assessment of the Tfh cell and germinal centre response in *Il2*^cre/+; *Rosa26*^stop-flox-Il2/+ transgenic mice that do not switch off IL-2 production, and *Il2*^cre/+; *Rosa26*^+/+ control mice 12 days after influenza A infection. Flow cytometric contour plots (**A**) and quantification of the percentage of CXCR5^highPD-1^highFoxp3^-CD4^+ Tfh cells in the mediastinal lymph node (**B**) and spleen (**C**). Flow cytometric contour plots (**D**) and quantification of the percentage of Bcl6^+Ki67^+B220^+ germinal centre B cells in the mediastinal lymph node (**E**) and spleen (**F**). The height of the bars indicates the median, each symbol represents one mouse, data are pooled from two independent experiments. p-Values calculated between genotype groups by Mann–Whitney *U* test.

infection, *Il2*^cre/+; *Rosa26*^stop-flox-Il2/+ mice have fewer Tfh cells in the draining lymph node and spleen (***Figure 8A–C***), which is associated with a reduced frequency of germinal centre B cells (***Figure 8D–F***). This provides a proof of concept that pro-inflammatory cytokine production needs to be limited to enable full Tfh cell differentiation in secondary lymphoid organs.

## Discussion

The formation of virus-neutralising antibodies after influenza vaccination provides protection against subsequent infection, yet the cellular and molecular pathways that support a high-titre antibody response in humans remain incompletely defined. Here a systems immunology approach was used to determine which immune parameters are associated with antibody formation upon vaccination. We used MHC class II tetramers to track the formation and differentiation of HA-specific CD4^+ T cells after vaccination, ensuring the specificity of the CD4^+ T cell response could be accurately matched to the A.Cali09 antibody response. HA-specific cTfh cell frequency was strongly correlated with anti-A. Cali09 antibodies 6 weeks after vaccination, and there was not a reproducible relationship between total HA-specific CD4^+ T cells and antibody titre, highlighting the importance of antigen-specific Tfh cell differentiation to support humoral immunity. TCR repertoire analysis showed that HA-specific cTfh cells formed from pre-existing memory cells in both younger and older adults, but their differentiation was reduced in older people. Interestingly, there was no difference in TCR repertoire diversity in cTfh cells in ageing, indicating that a contraction of the TCR repertoire with ageing is unlikely to be the cause of poor Tfh cell differentiation in older people. The defective cTfh response in ageing was, however, associated with an enhanced pro-inflammatory gene expression signature, suggesting that inflammation can limit the response to vaccination. These enhanced inflammatory signatures

associated with poor antibody titre in an independent cohort of influenza vaccinees. The dampening of Tfh cell formation by enhanced cytokine production was confirmed by the use of genetically modified mice where IL-2 production is restricted to the appropriate anatomical and cellular compartments, but once initiated cannot be inactivated. Together, this suggests that formation of antigen-specific Tfh cells is essential for high-titre antibody responses, and that excessive inflammatory factors can contribute to poor cTfh cell responses.

In order to understand what type of immune response supports protective antibody production upon vaccination, we comprehensively analysed the immune response to seasonal influenza vaccination by measuring IgG responses to 32 HA proteins, 8 cytokines and chemokines, B cells, CD4+ T cells, and HA-specific CD4+ T cells. This combined approach enabled us to identify which parameters were correlated with the antibody response after vaccination. In contrast to previous studies (*Bentebibel et al., 2013*; *Li et al., 2012*), we observed that the frequencies of circulating B cell populations including plasmablasts did not correlate with the long-term antibody response. This suggests that the circulating plasmablasts observed 7 days after vaccination in people are likely biomarkers of the early extrafollicular antibody response that provides a short initial burst of antibodies, rather than of long-lived plasma cells (*MacLennan et al., 2003*). However, the downregulation of BAFF and upregulation of one of its receptors, BCMA, in serum over the first 7 days after vaccination clearly demonstrate that dynamic regulation of B cell responses through BAFF and its receptors are tightly intertwined with the magnitude of the antibody output to vaccination. Consistent with this, the upregulation of *TNFRSF17*, the gene for BCMA, has been well reported to correlate with vaccine titres in PBMC transcriptomics studies (*Nakaya et al., 2015*; *Querec et al., 2009*; *Sobolev et al., 2016*; *Li et al., 2017*). Importantly, we show that expansion of cTfh cells, both polyclonal and HA-specific, was consistently correlated with the magnitude of antibody responses across two cohorts. Through sequencing the HA-specific cells, we were able to show that cTfh cells are recalled from the resting memory CD4+ T cell compartment, public TCR clonotypes are readily detectable in antigen-specific cTfh cells, and antigen-specific cTfh cells share a transcriptional program with lymph node Tfh that includes the downregulation of TNF and IL-2 signalling. As age is one of the key factors that influences antibody responses to vaccines (*Dugan et al., 2020*; *Frasca and Blomberg, 2020*; *Gustafson et al., 2020*), by clearly defining ideal immune responses in younger individuals we were able to gain novel insights into how ageing negatively impacts the immune response, resulting in low-titre antibody responses following seasonal influenza vaccination.

Impaired T cell responses to vaccines in ageing have been proposed to be caused by contraction of the TCR repertoire and the accumulation of terminally differentiated effector cells. However, in our study, we observed impaired antigen-specific Tfh differentiation in older people, despite no defect in the overall antigen-specific CD4+ T cell response. Through analysing the clonal relatedness in the TCR repertoire of HA peptide-specific T cells from before and after vaccination, we observed that similar frequencies of d7 cTfh cell clonotypes were recalled by vaccination from resting memory CD4+ T cell compartment in both age groups. This suggests that the ability of memory cells to be recalled by vaccines in ageing is not compromised. Furthermore, we observed no difference in the diversity of the TCR repertoire of the cTfh cells that formed after vaccination between age groups. Therefore, this suggests that a loss of diversity in the T cell repertoire with age does not explain the impaired Tfh differentiation we observed in our study. Impaired Tfh responses to vaccines have been observed in humans and mice (*Stebegg et al., 2020*; *Lefebvre et al., 2016*). Interestingly the formation of pre-Tfh cells remains intact, but there is a failure to generate bona fide GC Tfh cells (*Lefebvre et al., 2016*; *Webb et al., 2021*). Together with the data presented here, this suggests that instead of intrinsic defects in CD4+ T cells pre-vaccination in older people, the reduced cTfh cell frequency post-vaccination may be explained by extrinsic factors, such as inflammation, that lead to a failure to appropriately acquire the GC Tfh cell gene signature.

Ageing is often associated with a state of low-grade inflammation, termed 'inflammaging' (*Cunha et al., 2020*; *Pereira et al., 2020*; *McElhaney et al., 2020*). However, in this study we observed no age-related difference in inflammatory gene signatures in HA-specific CD4+ T cells at d0, which suggests that the cTfh cell response was negatively impacted by inflammation post-vaccination, rather than pre-existing inflammation having affected resting memory T cell function prior to vaccination. Because older people are more at risk of severe health outcomes after infection, significant effort has been made to alter vaccine formulations to enable them to be effective in this age group. Modifications

to vaccines to increase the antibody response in older individuals now include increasing the antigen dose and using more potent adjuvants (*DiazGranados et al., 2014*; *Squarcione et al., 2003*), which enhance the inflammatory response. We have previously demonstrated in older people and aged mice that type 1 interferon and TLR7 signalling is important for conventional dendritic cells type 2 to support Tfh differentiation after vaccination (*Stebegg et al., 2020*). These studies, together with the data presented here, prompt the hypothesis that while some types of inflammation are 'good' for promoting long-lived antibody responses in ageing, prolonged exposure to other cytokines such as IL-2 is 'bad' and negatively impacts Tfh cell differentiation. Consistent with this, inhibition of inflammatory monocyte recruitment into the skin of older people enhanced the local CD4 T cell response to varicella zoster virus antigen challenge (*Chambers et al., 2021*). Therefore, vaccination strategies that support humoral immunity through limiting 'bad' pro-inflammatory signalling may be the key to improving vaccine efficacy, particularly in older people.

Our data demonstrate that IL-2 is one signalling pathway that is normally downregulated during Tfh differentiation. IL-2 is a cytokine produced by T cells early after T cell activation that promotes clonal expansion and favours a Th1 cell fate at the expense of Tfh cell differentiation (*Ballesteros-Tato et al., 2012*; *Ray et al., 2015*), in line with the negative relationship observed here between cTfh cell differentiation and the response to IL-2 in those cells. Tfh cells are reported to be IL-2 producers as a consequence of heightened TCR signalling, and whilst they typically do not respond to IL-2, they provide a source of this cytokine to support non-Tfh cells (*DiToro et al., 2018*). Here, we have used genetically modified mice to demonstrate that enhanced IL-2 production in vivo results in fewer Tfh cells in secondary lymphoid organs, demonstrating that aberrantly produced IL-2 has negative consequences for Tfh cell formation.

TNF signalling in cTfh cells was also negatively correlated with Tfh differentiation and antibody response upon vaccination, and we found that it was enhanced in cTfh cells from older people consistent with a recent report (*Herati et al., 2021*). While this cytokine is necessary for the formation of primary B cell follicles and follicular dendritic cell networks that underpin the formation of germinal centres (*Tumanov et al., 2010*; *Pasparakis et al., 1996*; *Pasparakis et al., 2000*; *Wang et al., 2001*), it has also been implicated in the loss of germinal centres and disorganisation of secondary lymphoid organ architecture during infections and immunisations in mice (*Ryg-Cornejo et al., 2016*; *Popescu et al., 2019*). The role for TNF in Tfh cell differentiation or survival remains unclear, and it is noteworthy that this pleiotropic cytokine can regulate many aspects of T cell biology, including NF-$\kappa$B signalling, TCR signalling, and apoptosis pathways (*Mehta et al., 2018*). In our study, this TNF signalling signature did not include the *TNF* gene, nor were *TNF* transcripts altered in cTfh cells with age. While in our study serum concentrations for both TNF and IL-2 were below the limit of detection in most samples, serum TNF levels and TNF production from memory B cells have been reported to increase with age (*Carr et al., 2016*; *Frasca et al., 2014*), suggesting that our TNF signature in cTfh cells may result from paracrine sources . TNF/TNF-receptor signalling has been linked to reduced memory CD4[+] T cell frequencies after influenza infection in mice (*DeBerge et al., 2014*; *Singh et al., 2007*), and T cells from older people have been found to be more sensitive to TNF-alpha-induced apoptosis via the extrinsic pathway (*Gupta, 2002*; *Aggarwal et al., 1999*), suggesting that TNF may impact Tfh cell expansion or survival. In agreement with this, we observed some gene overlap between the apoptosis and TNF signatures downregulated in young cTfh cells on d7 relative to d0, and elevated in cTfh cells in older people. More research is required to understand how TNF may regulate germinal centre biology, particularly as elevated TNF correlated with impaired germinal centres and absence of Tfh cells observed in patients that succumbed to COVID-19 infection (*Kaneko et al., 2020*; *Gu et al., 2005*), and has been negatively correlated with Tfh cell frequency in symptomatic SARS-CoV2-infected patients (*Koutsakos et al., 2021*). Therefore, while it is unclear whether TNF acts directly on T cells to impair Tfh differentiation or survival, or has a broader effect on lymphoid organ architecture, our data indicate that persistence of TNF signalling in cTfh cells is linked to poor vaccine responses.

The need to develop effective vaccines that are effective in all age groups has been emphasised by the COVID19 pandemic, in which older people are more likely to become seriously ill, or die, after infection (*Verity et al., 2020*). Promisingly, multiple SARS-CoV2 vaccines are able to induce strong antibody responses in older people. While a single dose of adenoviral vectored vaccines results in a lower antibody titre in older people, this can be boosted to comparable levels to younger adults with a second dose (*Ramasamy et al., 2021*; *Zhu et al., 2020*). Mechanistic studies in mice show

that this is due to the second vaccine dose enhancing Tfh cells and the germinal centre response to a greater extent in aged mice than in younger adult animals (*Silva-Cayetano et al., 2021*). COVID19 mRNA vaccines induce high-titre antibodies after two doses in all age groups (*Widge, 2020*; *Prendecki et al., 2021*; *Collier et al., 2021*) and generate superior germinal centre responses than protein subunit vaccines in young mice (*Lederer et al., 2020*) and humans (pre-printed in *Ellebedy, 2021*). Together with the data presented here, this demonstrates that effective vaccines are ones that promote the germinal centre response. Therefore, vaccination strategies that induce the optimal inflammatory environment to support Tfh differentiation are key to generating enduring antibody-mediated immunity.

# Materials and methods

**Key resources table**

| Reagent type (species) or resource | Designation | Source or reference | Identifiers | Additional information |
|---|---|---|---|---|
| Other | Viability dye and labelling reagent | Thermo Fisher | Cat#: 65-0865-18 | Flow cytometry (1:10,000) |
| Antibody | Anti-human CD14 APC-eF780 clone: 61D3 (mouse monoclonal) | Thermo Fisher | Cat#: 47-0149-42 | Flow cytometry (0.5 µL/ million cells) |
| Antibody | Anti-human CD16 APC-eF780 clone: eBioCB16 (mouse monoclonal) | Thermo Fisher | Cat#: 47-0168-42 | Flow cytometry (0.3 uL/ million cells) |
| Antibody | Anti-human CD3 BV605 clone: UCHT1 (mouse monoclonal) | BioLegend | Cat#: 300460 | Flow cytometry (0.5 µL/million cells) |
| Antibody | Anti-human CD19 BB515 clone: HIB19 (mouse monoclonal) | BD Biosciences | Cat#: 564456 | Flow cytometry (1 µL/million cells) |
| Antibody | Anti-IgD BV421 clone: IA6-2 (mouse monoclonal) | BD Biosciences | Cat#: 563813 | Flow cytometry (1 µL/million cells) |
| Antibody | Anti-human CD38 APC clone: HIT2 (mouse monoclonal) | Thermo Fisher | Cat#: 17-0389-42 | Flow cytometry (0.5 µL/ million cells) |
| Antibody | Anti-human CD20 PECY7 clone: 2H7 (mouse monoclonal) | BioLegend | Cat#: 302312 | Flow cytometry (0.5 µL/ million cells) |
| Antibody | Anti-human CD27 BV650 clone: M-T271 (mouse monoclonal) | BD Biosciences | Cat#: 564894 | Flow cytometry (0.5 µL/million cells) |
| Antibody | Anti-human CD24 PerCP-eFluor 710 clone: eBioSN3 (mouse monoclonal) | Thermo Fisher | Cat#: 46-0247-42 | Flow cytometry (0.5 µL/million cells) |
| Antibody | Anti-human CD19 APC-eF780 clone: HIB19 (mouse monoclonal) | Thermo Fisher | Cat#: 47-0199-42 | Flow cytometry (0.3 µL/ million cells) |
| Antibody | Anti-human CD3 BUV 395 clone: UCHT1 (mouse monoclonal) | Thermo Fisher | Cat#: 563546 | Flow cytometry (0.5 µL/million cells) |
| Antibody | Anti-human CD45RA BUV737 clone: HI100 (mouse monoclonal) | BD Biosciences | Cat#: 564442 | Flow cytometry (0.5 µL/million cells) |
| Antibody | Anti-human CD4 PercpCy5.5 clone: RPA-T4 (mouse monoclonal) | BD Biosciences | Cat#: 560650 | Flow cytometry (0.3 µL/million cells) |
| Antibody | Anti-human CXCR5 BB515 clone: RF8B2 (rat monoclonal) | BD Biosciences | Cat#: 564624 | Flow cytometry (0.9 µL/ million cells) |
| Antibody | Anti-human PD1 APC clone: eBioJ105 (mouse monoclonal) | BD Biosciences | Cat#: 17-2799-42 | Flow cytometry (1 µL/million cells) |
| Antibody | Anti-human ICOS biotin clone: ISA-3 (mouse monoclonal) | Thermo Fisher | Cat#: 13-9948-82 | Flow cytometry (1 µL/million cells) |
| Antibody | Anti-human CXCR3 BV421 clone: 1C6/CXCR3 (mouse monoclonal) | Thermo Fisher | Cat#: 562558 | Flow cytometry (0.5 µL/million cells) |
| Antibody | Anti-human CCR6 BV786 clone: 11A9 (mouse monoclonal) | BD Biosciences | Cat#: 563704 | Flow cytometry (0.5 µL/million cells) |
| Other | Streptavidin BV650 | BD Biosciences | Cat#: 563855 | Flow cytometry (0.2 µL/ million cells) |

*Continued on next page*

*Continued*

| Reagent type (species) or resource | Designation | Source or reference | Identifiers | Additional information |
|---|---|---|---|---|
| Antibody | Anti-mouse CXCR5 clone: L138D7 (rat monoclonal) | BioLegend | Cat#:145511; RRID:AB_2562127 | Flow cytometry (1:50) |
| Antibody | Anti-mouse PD-1 clone: 29F.1A12 (rat monoclonal) | BioLegend | Cat#: 135231; RRID:AB_2566158 | Flow cytometry (1:5000) |
| Antibody | Anti-mouse Foxp3 clone: FJK-16s (rat monoclonal) | eBioscience | Cat#: 48-5773-82; RRID:AB_1518812 | Flow cytometry (1:400) |
| Antibody | Anti-mouse B220 clone: RA3-6B2 (Rat monoclonal) | eBioscience | Cat#: 15-0452-82; RRID:AB_468755 | Flow cytometry (1:500) |
| Antibody | Anti-mouse Ki67 clone: 16A8 (rat monoclonal) | BioLegend | Cat#: 652420; RRID:AB_2564285 | Flow cytometry (1:1000) |
| Antibody | Anti-mouse/human Bcl6 mouse clone: K112-91 (mouse monoclonal) | BD Biosciences | Cat#: 561525; RRID:AB_10898007B | Flow cytometry (1:200) |
| Antibody | Anti-mouse CD4 clone: GK1.5 (rat monoclonal) | BD Biosciences | Cat#: 563790; RRID:AB_2738426 | Flow cytometry (1:400) |

## Human cohorts

Peripheral blood was collected from healthy UK adults recruited through the NIHR Bioresource and vaccinated with the trivalent influenza vaccine. Participants were selected on the basis of having at least one allele of either *HLADR*0701* or *HLADR*1101* as determined by single-nucleotide polymorphism typing using the UK Biobank v2.1 Axiom array. Two cohorts were collected: cohort 1, Northern Hemisphere winter 2014–2015, n = 16 participants 18–36 years old, n = 18 participants 66–75 years old; cohort 2, Northern Hemisphere winter 2016–2017, n = 21 participants 18–36 years old, n = 21 participants 66–98 years old. Venous blood was collected for serum and into EDTA-coated tubes for PBMC processing on the day of vaccination (prior to administration of the vaccine), 7 and 42 days after vaccination. PBMCs were isolated by density gradient separation using Histopaque-1077 (Sigma), frozen in foetal bovine serum supplemented with 10% dimethyl sulfoxide (Sigma), and kept in liquid nitrogen prior to analysis by flow cytometry or flow sorting for RNA sequencing. Lymph node samples were taken from patients recruited from the renal transplant live donor program at Cambridge University Hospitals NHS Foundation Trust, as part of the routine operative procedure, as described previously (*Hill et al., 2019*; *Wallin et al., 2014*).

## Flu HA and Cytokine Luminex

IgG to influenza HA proteins were measured before and after vaccination by Luminex using magnetic beads coated with full-length recombinant HA proteins, as previously reported (*Wang et al., 2015*). Proteins were expressed using baculovirus expression system in insect cells and coupled to Bio-plex Pro Magnetic COOH Beads (Bio-Rad, Hercules, CA). Titres are represented as arbitrary units per mL, calculated from a total IgG standard curve. The lower limit of detection for each HA protein was defined as the mean + 2 standard deviations measured for the HA-coated beads with secondary antibody only. Where indicated, pre-existing IgG titres were subtracted to from day 7 and 42 titres to calculate vaccination-induced IgG responses. Serum samples were analysed for cytokine and chemokine concentrations using Human Magnetic Luminex kit (Cat# LXSAHM, R&D Systems) custom ordered to detect the following analytes: BAFF, CXCL13, BCMA, APRIL, TWEAK, Osteopontin, SCF, and Light. Analytes were included for further analysis if more than 50% of samples had values above the lower limit of detection.

## Haemagglutination inhibition assay

Antibody titres pre- and post-vaccination were determined using the haemagglutination inhibition (HAI) assay using the standard WHO protocol, as previously described (*Chen et al., 2010*). Sera were treated overnight with receptor-destroying enzyme (Denka Seiken Co.) and were subsequently tested by standard methods using four HA units of virus and a 0.5% suspension of turkey red blood cells. HAI titres were recorded as the reciprocal of the highest dilution of the serum that completely inhibited agglutination of erythrocytes by four HA units of the virus.

## Flow cytometry of B cells and FACS of T lymphocytes

Cryopreserved mononuclear cells were thawed and rested for 1 hr at 37°C. Fc receptors were blocked using anti-human CD32 antibody (clone 6C4, eBioscience). 4 million PBMCs were stained with a panel of antibodies to measure B cells (*Supplementary file 2*), acquired on BD LSR Fortessa5 cytometer, and gated according to the strategy outlined in *Figure 1—figure supplement 1*. For T cell staining and sorting, between 15 and 40 million PBMCs were first treated with 50 nM dasatinib (Sigma) for 10 min at 37°C, and then stained with PE-conjugated tetramers for 2 hr at room temperature with methods and reagents that have been previously reported (*Yang et al., 2013*; Benaroya Research Institute Tetramer Core Facility). Tetramers were loaded with peptides of HA protein specific to the A/California/04/2009(H1N1) influenza strain (GenBank: ACQ76318.1), with the HLADR*0701 tetramer containing the peptide ITFEATGNLVVPRYAFAMER, which corresponds to amino acids 257–276, and HLADR*1101 tetramer contained the peptide FYKNLIWLVKKGNSYPKLSK, which corresponds to amino acids 161–180 (*Yang et al., 2013*). Tetramer-stained PBMCs were then enriched for memory CD4$^+$ T cells by using magnetic separation (MagniSort enrichment, eBioscience), and subsequently stained with the antibody panel outlined in *Supplementary file 3*. HA-specific T cells were isolated using a FACSAria Fusion Sorter (BD Biosciences) by first excluding unwanted cell types using a dump channel consisting of viability dye and antibodies to CD14, CD16, and CD19. CD3+ CD4+ CD45RA- tetramer-binding cells were sorted and further phenotyping was performed using the gating strategy outlined in *Figure 1—figure supplement 2*. Up to 1000 HA-specific tetramer+ cells were sorted directly into 9 µL of lysis buffer containing RNAse inhibitor as per the SMART-Seq v4 Ultra Low Input RNA Kit for Sequencing manual (Takara), with the additional markers CXCR5 and PD1 used to isolate tetramer-binding cTfh cells in d7 samples. Additional CD4$^+$ T cell subsets were analysed post acquisition as per the gating strategy outlined in *Figure 1—figure supplement 3*.

## RNA sequencing

RNA sequencing and data processing mRNA was converted to cDNA using the SMART-Seq v4 Ultra Low Input RNA Kit (Takara Bio). Sequencing libraries were subsequently generated using the Nextera XT DNA Library Prep Kit (Illumina), followed by sequencing on the Illumina HiSeq 2000 with ~50 million 100 bp single-end reads per sample. mRNA from lymph node CD4$^+$CD45RA$^-$ T cell populations was isolated from 1000 cells sorted into lysis buffer from six individuals as previously described (*Hill et al., 2019*). Sequencing reads were aligned to the reference human genome GRCh38 using HISAT2 (*Kim et al., 2015*) and quantitated using Rsubread package (*Liao et al., 2019*). Samples were excluded on the basis of poor cDNA quality prior to sequencing, or where drop-out genes with zero counts represented more than 80% of total reads. Genes were filtered based on having more than 10 reads in 20% of samples, yielding 19,113 genes. Unwanted variation due to batch effects and sex was determined using the RUVseq package (*Risso et al., 2014*) for each cohort separately using the model.matrix [~data$sex+ data$group], where the 'group' factor represents the combination of time point, cell type, and age group. The RUVseq output was incorporated into DESeq2 model matrix design using the following model: [design = ~ $W\_1$+ $W\_2$+ $W\_3$+ sex + group], where $W$ represents the three RUVseq variance factors identified for each cohort. DESeq2 fold changes were adjusted using the lfcShrink normal method, and variance-stabilised normalisation was applied to the counts to give an expression value per gene (DESeq2 package; *Huber et al., 2002*; *Love et al., 2014*). Significantly differentially expressed genes had adjusted p-values of <0.1 and log$_2$ fold change of 0.5 or greater, with consistent changes across both cohorts or comparisons. Pathway analysis was performed using the Hallmark gene sets (MSigDB collections; *Liberzon et al., 2015*) on gene lists ranked by log$_2$ fold change using the fgsea R package (*Sergushichev, 2016*). Significantly enriched pathways had adjusted p-values of <0.1 and normalised enrichment scores of <–1 or >1, consistently across both cohorts or comparisons. The lymph node germinal centre Tfh signature was generated by comparing Tfh with CD4$^+$ CD45RA-CXCR5- and CD4$^+$ CD45RA-CXCR5+ PD1- non-Tfh cell populations. Genes were selected on the basis of a DESeq2 adjusted p-value cut-off <0.1 and log$_2$ fold change of greater than or less than 0.5, for both non-Tfh populations. Pathways analysis using the fgsea package was used to define enriched pathways as described above.

Signature genes were determined from the fgsea-defined leading-edge gene lists for TNF signalling pathway, IL-2 STAT5 signalling pathway, and inflammation pathways (Interferon gamma, Inflammatory response, and Complement combined). Genes present in the leading edge for both cohorts were

included, and where a gene was present across multiple pathways, it was assigned to the pathway where it was ranked highest, ensuring each gene signature was non-overlapping. VSN-normalised gene expression values for selected signatures were then converted to z-scores with all time points combined, and then z-scores were summed for each sample (*Lee et al., 2008*). Microarray data of PBMCs from GSE74813 (*Nakaya et al., 2015*; *Franco et al., 2013*) were re-analysed as follows: raw .CEL files were downloaded from GEO with the corresponding annotation data. CEL files were read into R via Affy package (*Gautier et al., 2004*) and were normalised using VSN (*Huber et al., 2002*). The dataset was then filtered to select only individuals with paired d0 and d7 samples. Gene z-scores were then calculated and summed across gene signatures for each sample as described above, with the summed gene signature scores for d0 subtracted from d7 scores for each sample for correlation to antibody titres, which were made available upon request (*Nakaya et al., 2015*).

## TCR sequencing and clonotype analysis

TCRβ clonotypes were called from adaptor-trimmed RNA sequencing fastq files using MIXCR (version 2.1.9; *Bolotin et al., 2015*) run in RNA-Seqmode with rescuing of partial alignments and set to collate *TCRA* or *TCRB* clonotypes at the amino acid level and requiring more than five reads to identify a clonotype. Clonotype diversity was determined using vdjtools (version 1.0.3; *Shugay et al., 2015*). Recalled *TCRB* clones were determined by analysing paired d0 and d7 samples from the same individual for both cohorts combined, and clonotype defined by CDR3 amino acid sequence, V, D, and J gene usage. Public *TCRB* clonotypes were analysed for both cohorts combined, and each time point separately, with V and J gene usage among public clones compared for each time point and genotype separately.

## Vaccination trajectory and pseudotime analysis

The trajectory analysis was assembled as previously described (*Hill et al., 2020*; *Alpert et al., 2019*). Briefly, the frequencies for immune parameters were first normalised for abundance by subtracting the mean and dividing by the standard deviation for both cohorts combined. Variables with missing values were excluded, leaving 35 variables analysed for each time point (*Supplementary file 1*) PC analysis was then performed, in which the samples from different time points showed separated by PC1. Therefore, the immune parameters that had a correlation to PC1 greater than 0.4 were included (n = 23 cell types). The diffusion maps algorithm was then applied to the scaled frequencies using the destiny R package (*Angerer et al., 2016*; *Haghverdi et al., 2016*), the resulting diffusion pseudotime values (described as vaccination trajectory) were scaled to a range of 0–1 and compared between time points and age groups. Correlations between vaccination trajectory and cell-type frequencies were analysed by Spearman's correlation.

## Influenza A infection of genetically modified mice

*Il2*cre (*Yamamoto et al., 2013*) and *Rosa26*stop-flox-Il2 [37] mice were bred and maintained in the Babraham Institute Biological Support Unit. No primary pathogens or additional agents listed in the Federation of European Laboratory Animal Science Association recommendations were detected during health monitoring surveys of the stock-holding rooms. Ambient temperature was ~19–21°C and relative humidity 52% . Lighting was provided on a 12 hr light:12 hr dark cycle including 15 min 'dawn' and 'dusk' periods of subdued lighting. After weaning, mice were transferred to individually ventilated cages with 1–5 mice per cage. Mice were fed Mouse Breeder and Grower CRM (P) VP diet (Special Diet Services) ad libitum and received seeds (e.g. sunflower, millet) at the time of cage-cleaning as part of their environmental enrichment. All mouse experimentation was approved by the Babraham Institute Animal Welfare and Ethical Review Body. Animal husbandry and experimentation complied with existing European Union and United Kingdom Home Office legislation and local standards (PPL: PP3981824). Influenza infections were done as previously described (Denton JEM 2019), and mice were administered $10^4$ plaque-forming units of influenza A/Hong Kong/1/1968/x31 virus intranasally under inhalation anaesthesia.

## Analysis of mouse lymphocytes by flow cytometry

Tissue processing and staining was performed as previously described (*Whyte, 2020*). Single-cell suspensions were generated by disrupting spleens and lymph nodes with glass slides before being

filtered through 100 μm mesh. Cells were counted using a Countess cell counter (Thermo Fisher). Approximately 2 million cells were stained with flow cytometry antibodies. Non-specific binding was blocked using 2.4G2 supernatant and dead cells were labelled by fixable viability dye eFluor 780 (Thermo Fisher). Single-cell suspensions were stained with the following antibodies: anti-CXCR5 clone L138D7, anti-CD4 clone GK1.5, anti-PD-1 clone 29F.1A12, anti-Foxp3 clone FJK-16s, anti-B220 clone RA3-6B2, anti-Ki67 clone 16A8, and anti-Bcl6 clone K112-91. For intracellular staining, cells were fixed and permeabilised the Foxp3 Transcription Factor Staining Buffer Set (eBioscience) according to the manufacturer's instructions. Flow cytometry samples were acquired on a Yeti/ZE5 (Propel Labs/ Bio-Rad) or Aurora (Cytek) spectral flow cytometer. Data was compensated via AutoSpill (*Roca et al., 2021*).

## Statistics

All statistical tests for cell-type frequencies assumed non-parametric data. Fold change for immune parameters was calculated by dividing d7 or d42 frequency by d0, and post-vaccination increases relative to baseline were determined by subtracting d0 from d7 or d42 data. Two-group comparisons were made using either two-tailed Mann–Whitney tests or Wilcoxon tests for paired data from the same individual at different time points. Multiple group comparisons p-values were calculated using Dunn's post hoc test. Correlation analyses used Spearman correlation, with the exception of log-normalised TCRB clone frequency analysis, which used Pearson's correlation. As necessary, false discovery rate (FDR)(Benjamini–Hochberg) adjustment was used to adjust for multiple testing on two-group comparisons. Heatmaps of manual gated cell subsets that were altered by vaccination or age group were generated using Pheatmap package (*Kolde, 2018*).

## Acknowledgements

The NIHR Cambridge Biomedical Research Center (BRC) is a partnership between Cambridge University Hospitals NHS Foundation Trust and the University of Cambridge, funded by the National Institute for Health Research (NIHR). We are indebted to the NIHR Cambridge BRC volunteers for their participation and thank the NIHR Cambridge BRC staff for their contribution in coordinating the vaccinations and venepuncture. We thank the staff of the Babraham Institute Flow Cytometry Facility and the Sequencing Facility for their technical assistance. We are grateful to the Babraham Bioinformatics Group for their help. This study was supported by H2020 European Research Council funding awarded to MAL (637801-TWILIGHT) and the Biotechnology and Biological Sciences Research Council (BBS/E/B/000C0427, BBS/E/B/000C0428, and the Campus Capability Core Grant to the Babraham Institute). DLH is supported by a National Health and Medical Research Council Australia Early-Career Fellowship (APP1139911). JCL is supported by a Wellcome Trust Intermediate Clinical Fellowship (105920/Z/14/Z). MAL is an EMBO Young Investigator and Lister Institute Prize Fellow. JLL is supported by a National Science Scholarship (PhD) from the Agency for Science, Technology and Research, Singapore. This paper presents independent research supported by the NIHR Cambridge BRC. The views expressed are those of the authors and not necessarily those of the NIHR or the Department of Health and Social Care.

## Additional information

### Funding

| Funder | Grant reference number | Author |
| --- | --- | --- |
| Biotechnology and Biological Sciences Research Council | BBS/E/B/000C0427 | Michelle A Linterman |
| Biotechnology and Biological Sciences Research Council | BBS/E/B/000C0428 | Michelle A Linterman |
| National Health and Medical Research Council | APP1139911 | Danika L Hill |

| Funder | Grant reference number | Author |
|---|---|---|
| Wellcome Trust | 105920/Z/14/Z | James C Lee |
| Agency for Science, Technology and Research | PhD Scholarship | Jia Le Lee |
| Lister Institute of Preventive Medicine | Lister Fellowship | Michelle A Linterman |

The funders had no role in study design, data collection and interpretation, or the decision to submit the work for publication.

## Author contributions

Danika L Hill, Conceptualization, Data curation, Formal analysis, Investigation, Methodology, Project administration, Visualization, Writing - original draft, Writing - review and editing; Carly E Whyte, James Dooley, Formal analysis, Investigation, Methodology, Writing - review and editing; Silvia Innocentin, Jiong Wang, Martin S Zand, Investigation, Methodology; Jia Le Lee, Formal analysis, Investigation, Writing - review and editing; Eddie A James, William W Kwok, Methodology; James C Lee, Investigation; Adrian Liston, Methodology, Supervision, Writing - review and editing; Edward J Carr, Conceptualization, Data curation, Investigation, Methodology, Project administration; Michelle A Linterman, Conceptualization, Funding acquisition, Investigation, Methodology, Project administration, Supervision, Writing - original draft, Writing - review and editing

## Author ORCIDs
Danika L Hill ⑩ http://orcid.org/0000-0001-6284-7061
Edward J Carr ⑩ http://orcid.org/0000-0001-9343-4593
Michelle A Linterman ⑩ http://orcid.org/0000-0001-6047-1996

## Ethics

Human subjects: All human blood and tissue were collected in accordance with the latest revision of the Declaration of Helsinki and the Guidelines for Good Clinical Practice (ICH-GCP). The seasonal UK influenza vaccination cohort was collected with UK local research ethics committee approval (REC reference 14/SC/1077), using the facilities of the Cambridge Bioresource (REC reference 04/Q0108/44). Lymph node samples were collected from UK adults undergoing surgery for their own medical care, under ethical approval from UK Health Research Authority (REC reference 11/EE/0355, respectively), at Cambridge University Hospitals, and processed at the Babraham Institute. Written informed consent was received from all volunteers.

All mouse experimentation was approved by the Babraham Institute Animal Welfare and Ethical Review Body. Animal husbandry and experimentation complied with existing European Union and United Kingdom Home Office legislation and local standards (PPL: PP3981824).

## Decision letter and Author response
Decision letter https://doi.org/10.7554/eLife.70554.sa1
Author response https://doi.org/10.7554/eLife.70554.sa2

# Additional files

## Supplementary files
• Supplementary file 1. Immunological variables. Table of the immunological parameters used in this study

• Supplementary file 2. Inflammation, TNF, and IL-2 gene signatures. Table of the gene names used for the inflammation, TNF, and IL-2 gene signatures.

• Supplementary file 3. Antibody panel for B cells. Table of the flow cytometry staining panel used for identifying B cell subsets.

• Supplementary file 4. Antibody panel for T cells. Table of the flow cytometry staining panel used for identifying T cell subsets.

• Transparent reporting form

• Source data 1. 18–36-year-old samples d7 cTfh vs. d0 gene list.

- Source data 2. Lymph node germinal centre Tfh gene signature.
- Source data 3. d7 Tfh gene set.
- Source data 4. d7 vaccination genes.
- Source data 5. Ageing d7 cTfh vs. d0 gene list.

### Data availability

Sequencing data have been desposited in GEO accession GSE176447. All data analysed during this study are included in this manuscript or supporting files or have been published on Zenodo.

The following dataset was generated:

| Author(s) | Year | Dataset title | Dataset URL | Database and Identifier |
|---|---|---|---|---|
| Hill DL, Linterman MA | 2021 | RNA-sequencing of Influenza A.Cali09 specific CD4+ T cells from healthy UK adults | https://www.ncbi.nlm.nih.gov/geo/query/acc.cgi?acc=GSE176447 | NCBI Gene Expression Omnibus, GSE176447 |
| Hill DL, Linterman MA | 2021 | lintermanlab/Hill_influenza_Tfh_Ab_responses | https://doi.org/10.5281/zenodo.5606553 | Zenodo, 10.5281/zenodo.5606553 |

The following previously published datasets were used:

| Author(s) | Year | Dataset title | Dataset URL | Database and Identifier |
|---|---|---|---|---|
| Nakaya HI, Pulendran B | 2015 | Time Course of Adults Vaccinated with Influenza TIV Vaccine during 2010/11 Flu Season (HIPC cohort) | https://www.ncbi.nlm.nih.gov/geo/query/acc.cgi?acc=GSE74813 | NCBI Gene Expression Omnibus, GSE74813 |
| Linterman M, Pierson W, Hill D, Carr E, Wingett S | 2019 | RNA-seq of circulating Tfh like cells at day zero and seven and 28 relative to experimental vaccination dosing | https://www.ncbi.nlm.nih.gov/geo/query/acc.cgi | NCBI Gene Expression Omnibus, GSE131088 |

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
