## [Decision Letter]

**Acceptance summary:**

This paper will be of significant interest to immunologists interested in understanding the determinants of antibody responses to vaccination. It uses tetramers to identify and track CD4^+^ T cell responses to influenza vaccination in younger and older adults, in an effort to understand why older individuals tend to have reduced antibody responses to immunization.

**Decision letter after peer review:**

Thank you for submitting your article "Impaired HA-specific T follicular helper cell and antibody responses to influenza vaccination are linked to inflammation in humans" for consideration by *eLife*. Your article has been reviewed by 2 peer reviewers, and the evaluation has been overseen by a Reviewing Editor and Betty Diamond as the Senior Editor. The reviewers have opted to remain anonymous.

Essential revisions:

The reviewers and editor were in consensus about the necessary revisions, which are listed separately under each review below. Many involve changes to the framing or interpretation of results, but some additional data (likely easily obtainable) is also requested.

*Reviewer #2 (Recommendations for the authors):*

Overall, I am very enthusiastic about this manuscript, but feel the data analysis and discussion could be strengthened in several key areas:

1) Comparison of the d7 tet+ cTFH vs d0 CD4 tet+ populations (Figure 4) provides useful information regarding genes involved in cTFH differentiation, but it would be equally informative to assess the differences in d7 tet+ cTFH and d7 tet+ CXCR5- cells, to further understand how recall of responses into the cTFH pool differs from responses lacking a cTFH phenotype. At present, it is difficult to know whether the downregulation of inflammatory signalling pathways is specific to the cTFH compartment, or is shared by all recalled tet+ cells.

2) Relating to point #1 raised in the public review, Figure 5H does indicate that in both the overall data set and in cohort 1, there is a significant reduction in total tet+ cells (as % of CD4) in the older cohort compared to the younger group. The authors generally seem to minimise this point in order to focus on the cTFH population, but it is unclear to what extent there is a general defect in HA-specific T cell recall/expansion in older individuals.

3) My major concern is around the discussion of the implications of the inflammation-related gene sets that are differentially expressed among cTFH in older individuals. Without additional data, it seems difficult to me to conclude that vaccine-induced inflammation is driving poor cTFH differentiation, as the authors have not shown any such causative relationship. It seems equally plausible that a defect in antigen presentation, DC function, etc that impacts on initial CD4 Tmem activation and subsequent differentiation could cause the observed cTFH defect, which is simply then reflected in transcriptional pathways that are associated with the cTFH phenotype.

---

## [Author Response]

Essential revisions:The reviewers and editor were in consensus about the necessary revisions, which are listed separately under each review below. Many involve changes to the framing or interpretation of results, but some additional data (likely easily obtainable) is also requested.Reviewer #2 (Recommendations for the authors):Overall, I am very enthusiastic about this manuscript, but feel the data analysis and discussion could be strengthened in several key areas:

Thank you for the positive evaluation of our manuscript, and the opportunity to improve the paper in response to these comments.

1) Comparison of the d7 tet+ cTFH vs d0 CD4 tet+ populations (Figure 4) provides useful information regarding genes involved in cTFH differentiation, but it would be equally informative to assess the differences in d7 tet+ cTFH and d7 tet+ CXCR5- cells, to further understand how recall of responses into the cTFH pool differs from responses lacking a cTFH phenotype. At present, it is difficult to know whether the downregulation of inflammatory signalling pathways is specific to the cTFH compartment, or is shared by all recalled tet+ cells.

To determine whether the failure to downregulate of the inflammatory signalling pathways in ageing is specific to the cTfh cell compartment, or is shared by all recalled Tet+ cells at day 7 we have looked at these gene signatures in all Tet+ cells that were sequenced in the second cohort of vaccinees. This data indicates that the persistence of the TNFA signature in older people is specific to cTfh cells, but that the enhanced IL-2 signature is present in all Tet+ cells. These data have been added to Figure 6H and have been included in the results text on pages 10-11.

Results, pages 10-11:

“To further resolve these age-related transcriptional differences, we performed enrichment analysis with the Hallmarks genesets and observed 7 gene sets that were consistently positively enriched in d7 HA-specific Tet+cTfh cells from over 65 year old compared to 18-36 year old individuals (Figure 6H). Four of these elevated genesets (TNFA signalling via NFKB; IL2 STAT5 signalling; Apoptosis; Hypoxia) we previously identified as negatively enriched in Tet+cTfh from 18-36 year old individuals compared to d0 Tet+ cells. Of these enriched gene sets, all except TNFA signalling via NFKB were also enriched in all Tet+ cells in older persons in addition to cTfh cells, suggesting that there are both common and unique pathways for how vaccine-reactive CD4+ cells and cTfh cells sense their environment in ageing. This indicates that these genes are normally downregulated as HA-specific CD4+ T cells differentiate into Tfh cells in response to vaccination, and that this process appears dysregulated in older individuals. The remaining three enriched genesets (Interferon γ response; Inflammatory response; Complement) are suggestive of heightened pro-inflammatory responses in cTfh cells, but not in all Tet+ CD4 T cells, from older individuals, and included enrichment for IL6, CCL2 and CCL5.”

2) Relating to point #1 raised in the public review, Figure 5H does indicate that in both the overall data set and in cohort 1, there is a significant reduction in total tet+ cells (as % of CD4) in the older cohort compared to the younger group. The authors generally seem to minimise this point in order to focus on the cTFH population, but it is unclear to what extent there is a general defect in HA-specific T cell recall/expansion in older individuals.

In the first cohort we observed a statistically significant reduction in the frequency of Tet+ cells in older persons compared to younger individuals. However, as this did not replicate in the second cohort we did not feel able to draw strong conclusions about whether the total Tet+ population is diminished by ageing, which is why it was not a focus of the paper. But, it is important to establish whether there is a relationship between the degree of Tet+ CD4+ T cell expansion and cTfh cell differentiation with age, to do this we performed correlation analyses. There is no correlation between the expansion of Tet+ cells and the frequency of cTfh cells formed seven days after immunisation in either age group. This suggests that the impaired cTfh cell differentiation in older persons is most likely caused by factors other than the capacity of CD4+ T cells to expand after vaccination. These data have been added as Figure 5—figure supplement 1D, and included in the results text on page 8.

Text changes, Page 8:

“There was no consistent difference in the total d7 Tet+ HA-specific T cell population with age for both cohorts (Figure 5H) and we observed no age-related correlation between the ability of an individual to differentiate Tet+ cells into a cTfh cell and the overall expansion of Tet+ HA-specific T cell population (Figure 5—figure supplement 1D). Thus, our data suggests that the poor vaccine antibody responses in older individuals is impacted by impaired cTfh cell differentiation (Figure 5J) rather than size of the vaccine-specific CD4+ T cell pool.”

3) My major concern is around the discussion of the implications of the inflammation-related gene sets that are differentially expressed among cTFH in older individuals. Without additional data, it seems difficult to me to conclude that vaccine-induced inflammation is driving poor cTFH differentiation, as the authors have not shown any such causative relationship. It seems equally plausible that a defect in antigen presentation, DC function, etc that impacts on initial CD4 Tmem activation and subsequent differentiation could cause the observed cTFH defect, which is simply then reflected in transcriptional pathways that are associated with the cTFH phenotype.

We agree with the reviewer that the data presented in figure 7 are correlative, rather than causative. In order to test the hypothesis that increased inflammatory cytokine production in lymphoid organs limits Tfh cell differentiation we have used Il2cre/+; Rosa26stop-flox-Il2/+ transgenic mice. In this mouse model, IL-2-dependent cre-recombinase activity facilitates the expression of low levels of IL-2 in cells that have previously expressed IL-2. This creates a scenario in which cells that physiologically express IL-2 cannot turn off its expression therefore increasing expression IL-2 after antigenic stimulation (Whyte et al., bioRxiv, 2020, doi: https://doi.org/10.1101/2020.12.18.423431). After influenza infection, Il2cre/+; Rosa26stop-flox-Il2/+ transgenic mice have fewer Tfh cells in the draining mediastinal lymph node and in the spleen (Figure 8A-C), this is accompanied by a reduction in the magnitude of the GC B cell response (Figure 8D-E). These data provide a proof of concept that increased inflammatory signals in lymphoid organs limit the formation of Tfh cells, consistent with the negative correlation of IL-2 signalling and cTfh cell formation in humans (Figure. 7). This new data supports the conclusion that excess IL-2 signalling can limit the Tfh cell response. These data are presented in Figure 8, and are discussed on page 12 in the results, and pages 12-13 in the discussion.

We also agree with the reviewer that ageing is complex, and it is likely that there are multiple contributing factors to the poor immune response upon vaccination in older people. Here, we identify IL-2 signalling as one such factor, but have revised our discussion on pages 14-15 to clearly communicate that other pathways/cell types have also been implicated in ageing, and the causes are likely multi-factorial.

Results text, page 12.

“Sustained IL-2 production inhibits Tfh cell frequency and the germinal centre response. To test the hypothesis that cytokine signalling needs to be curtailed to facilitate Tfh cell differentiation turned to a genetically modified mouse model in which cells that have initiated IL-2 production cannot switch it off, Il2cre/+; Rosa26stop-flox-Il2/+ mice (37). Twelve days after influenza infection Il2cre/+; Rosa26stop-flox-Il2/+ mice have fewer Tfh cells in the draining lymph node and spleen (Figure 8A-C), which is associated with a reduced frequency of germinal center B cells (Figure 8D-F). This provides a proof of concept that proinflammatory cytokine production needs to be limited to enable full Tfh cell differentiation in secondary lymphoid organs.”

Discussion text, pages 12, 13:

These enhanced inflammatory signatures associated with poor antibody titre in an independent cohort of influenza vaccinees. The dampening of Tfh cell formation by enhanced cytokine production was confirmed by the use of genetically modified mice where IL-2 production is restricted to the appropriate anatomical and cellular compartments, but once initiated cannot be inactivated Together, this suggests that formation of antigen-specific Tfh cells is essential for high titre antibody responses, and that excessive inflammatory factors can contribute to poor cTfh cell responses.”

Discussion text, pages 14, 15:

“We have previously demonstrated in older people and aged mice that type 1 interferon and TLR7 signalling is important for conventional dendritic cells type 2 to support Tfh differentiation after vaccination (9). These studies, together with the data presented here, prompt the hypothesis that while some types inflammation are ‘good’ for promoting long-lived antibody responses in ageing, prolonged exposure to other cytokines such IL-2 are ‘bad’ and negatively impact Tfh cell differentiation.”